# Physics-based parameterisation framework for basal melting in ice-ocean boundary layers over dynamically stable pycnoclines
T. Jayasankar [1,2] & A. Jenkins [1] ✉

Accurate basal melt prediction is crucial for assessing ice sheet stability and sea level rise. Recent observations at eastern Thwaites Glacier reported low melt rates despite warm ocean waters. Weak vertical mixing due to low current speeds and strong density stratification suppresses melting. However, the basal melt parameterization approach in ocean models overestimates the melt rates there. Hence, we revisit the parameterization by applying an ice-ocean boundary current model to a simple horizontal ice base. This setting creates a boundary layer (BL) over a dynamically stable pycnocline. We show that the pycnocline's low diffusivity restricts heat transfer, causing models to overpredict melting, especially for weaker far-field currents. While reducing the prescribed BL depth can minimize this overprediction in ocean models, a better fix might be prescribing an upper melt rate limit for slower currents. We also propose a physics-based parameterization framework that more accurately emulates physics in models and observations.

Floating ice shelves are an integral part of the climate system as they regulate the outflow from the Antarctic Ice Sheet. Their thinning can result in the acceleration of upstream ice flow and the retreat of grounding lines[1]. Basal melting under the ice shelves is not only influenced by ocean forcing but also by the geometry of the ocean cavity beneath the ice[2–4]. Considering that melting changes the geometry of ice shelves and the dependence of melting on geometry, accurate basal melting in coupled ice-sheet ocean models is important for capturing this important feedback mechanism.

The relationship between melting and geometry is primarily governed by the depth-dependence of the in-situ freezing point ($T_f$) of ocean water. The freezing point at the ice-ocean interface (IOI) is usually expressed as a function of the depth (z) dependent pressure (P) and salinity (S) as follows, where $\lambda_1$, $\lambda_2$, and $\lambda_3$ are constants:

$$T_f = \lambda_1 S + \lambda_2 + \lambda_3 P. \qquad (1)$$

Ocean driven melting is governed by the difference between the ocean temperature (T) and the freezing point at the IOI ($T_f$), referred to as the ocean thermal driving ($T_*$). While the ambient thermal driving ($T_{*a}$), which is the difference between far field ocean temperature ($T_a$) and $T_f$ at the IOI, ultimately drives the process, the thermal driving near the IOI can

be considerably lower because of mixing between warm ambient ocean water and cold basal meltwater produced at the IOI. The resulting heat flux from ocean to ice and the consequent melting ($\dot{m}$) is generally expressed[5] as a linear function of $T_*$ near the IOI, a friction velocity ($u_*$) and a turbulent transfer coefficient ($\Gamma$):

$$\dot{m} = \left(\frac{c}{L_i}\right) u_* \Gamma T_*, \qquad (2)$$

Equation 2 is expressed in its reduced form ignoring the conductive heat flux into the ice. Basal melt parameterisations are commonly built on Eq. 2, where the underlying physics of ice-ocean heat transfer is addressed.

Recent observations from under the eastern Thwaites Glacier grounding zone[6,7] show low melt rates (2-5 m yr$^{-1}$) despite warm ambient ocean conditions. That result has been attributed to low current speeds and strong density stratification under the IOI suppressing the vertical mixing of heat towards the IOI. The canonical model of ice-shelf basal melting overestimates the melt rates (14 to 30 m yr$^{-1}$) under those conditions. While model studies[8,9] suggest some overprediction of $\dot{m}$ to be common, the scale for Thwaites is concerning. Hence, we revisit basal melt parameterisations in

[1]University of Northumbria, Newcastle upon Tyne, UK. [2]Now at Norwegian Polar Institute Fram Centre, Tromsø, Norway.
✉e-mail: adrian2.jenkins@northumbria.ac.uk

ocean models for cases in which stable stratification suppresses the vertical mixing.

For z-coordinate ocean models, when $T_*$ is evaluated at the nearest grid point to the IOI, the computed melt rates ($\dot{m}$) contain artefacts caused by the varying distance between the uppermost grid point and the sloped IOI. This shortcoming is circumvented by prescribing a boundary layer (BL) of uniform depth (h), a strategy[8] that has been widely followed[9–11]. A BL is a layer of fluid close to a solid boundary (the IOI in this case) where the effect of the boundary is felt by the fluid in motion. Within the BL the frictional drag or resistance created by the boundary is substantial[12]. There are velocity gradients in the direction perpendicular to the boundary, that gradually disappear when moving away from the boundary. The velocity gradients promote turbulent mixing of fluid particles across the BL. Beyond the BL, the flow shows negligible or zero influence from the boundary. Hence, the heat fluxes are expressed as a function of both currents and $T_*$ averaged over a prescribed BL of depth h[8]. Mathematically, melt rate $\dot{m}_h$ in this BL model (BLM) can be derived using the following equations:

$$u_{*h} = \frac{1}{2}\sqrt{C_d}\left|\frac{1}{h}\int_{z_b-h}^{z_b} u(z)dz\right|, \tag{3}$$

$$T_{*h} = \frac{1}{h}\int_{z_b-h}^{z_b} T_*(z)dz, \tag{4}$$

$$\dot{m}_h = \left(\frac{c}{L_i}\right)u_{*h}.\Gamma.T_{*h}. \tag{5}$$

In Eq. 3, $C_d$ denotes a drag coefficient and $z_b$ denotes the depth of the ice shelf base or the IOI, while $u(z)$ denotes the profile of currents under the IOI. In the actual BLM[8] both T and S are treated as separate variables resulting in the expansion of Eq. 5 to two separate equations for the BL heat and salt fluxes. The reduced equation using depth averaged BL thermal driving ($T_{*h}$) makes it easier to draw parallels in the following analysis while preserving the fundamentals.

Unlike for a steep IOI, for a nearly flat IOI (as is the case beneath Thwaites Eastern Ice Shelf), the buoyant, cold meltwater formed at the IOI cannot rise away from its source region. Buoyancy traps it between the IOI and the warmer saltier waters below. Mixing due to BL turbulence causes the meltwater to spread away from the IOI and can result in the formation of a relatively well-mixed layer of cold, buoyant waters above a sharp transition (the pycnocline) to the warmer and saltier ocean waters below. Considering the well-known nature of such a mixed layer, which parameterisations such as that of BLM are designed to capture, the reported mismatch in observed and calculated melt rates warrants an investigation. Here, we use an ice-ocean boundary current model[13] (IOBCM, see methods), which can reproduce both stratification due to melting and mixing due to ocean currents. We compare the actual model $\dot{m}$ with the $\dot{m}_h$ produced when the BLM parameterisation is applied to the IOBCM results. We study the BL dynamics in the IOBCM to diagnose and address the causes of the mismatch.

## Results
### Boundary layer dynamics
We first investigate BL dynamics for a horizontal IOI using the IOBCM with two different mixing schemes. This is to demonstrate that regardless of the mixing scheme used, the IOBCM yields qualitatively similar results. The mixing schemes used (see Methods) are the classic Pacanowski and Philander (PP) and a hybrid turbulence closure (HTC). A set of IOBCM simulations (see dataset-1 in Table 1) with varying model parameters give a like-for-like comparison of these mixing schemes. Unless specified otherwise, in all subsequent discussions of the IOBCM, the mixing scheme used is HTC owing to its more advanced physics[13].

From the BL dynamics in both schemes (Fig. 1a, b, c, d), mixing due to turbulence spreads the buoyant, colder meltwater produced at the IOI to

deeper levels, resulting in an upper layer with relatively uniform properties. We consider this well mixed layer as the BL, noting that its viscosity-dependent depth is consistent with that of a planetary boundary layer, beyond which the currents are close to geostrophic balance. Hence, based on the profile of $T_*$ (Fig. 1d), we define the BL depth, $D_{BL}$, as the depth where $\left|\frac{d^2 T_*}{dz^2}\right|$ is maximum. In a more turbulent BL, the effective viscosity is higher and the buoyant, cold waters are spread over greater depths (Fig. 1a, b, c, d, profiles in red for a far field current speed, $U_{FF} \cong 0.35$ m s$^{-1}$). Conversely, weaker turbulence leads to weaker mixing and buoyant, cold waters are confined closer to the ice shelf base (Fig. 1a, b, c, d, profiles in blue for $U_{FF} \cong 0.14$ m s$^{-1}$). The structure found with both mixing schemes is qualitatively similar, conforming to a BL over a dynamically stable pycnocline (BL-DSP) separating the buoyant, colder waters of the BL from the warmer, denser far-field ocean, albeit with differing BL depths determined by the differing viscosities determined by the individual mixing schemes.

The quantitative differences between the schemes, as mentioned above, are caused by the respective diffusivities (K) and viscosities ($\vartheta$), which are lower in the PP scheme than in HTC. Consequently, velocity gradients (Fig. 1a, b) in the HTC scheme are greater leading to higher values of friction velocity ($u_{*0}$) at the IOI (Fig. 1e) for any given far field current ($U_{FF}$). Two trajectories of $u_{*0}$ against $U_{FF}$ are shown in Fig. 1e for each scheme, corresponding to the two different values of $\sqrt{C_d}$ used (0.05 and 0.12) in the simulations, while the boundary layer thickness is determined almost uniquely by $u_{*0}$ regardless of the chosen drag coefficient or mixing scheme (Fig. 1f). Finally, we look at the effective thermal driving ($T_{*\delta}$), which is the thermal driving at a small distance $\delta$ ($< 0.5$m) from the IOI, as this governs the melt rates at the IOI. In Fig. 1g, h, the thermal driving ratio, $T_{*\delta}/T_{*a}$, is plotted against log ($u_{*0}$). Ice-ocean heat exchange becomes more efficient with increasing $u_{*0}$ (Eq. 24 in Methods) enhancing the melt rates but trapping the melted waters within the BL. While the growth of the BL with $u_{*0}$ increases its thermal inertia, the increasing heat loss at the IOI with $u_{*0}$ cools the BL. Hence, though counter intuitive, in both schemes, with an increase in $u_{*0}$, $T_{*\delta}$ decreases. Cooling of the BL with increasing currents has been reported in modelling studies[14] although not explicitly attributed to fluxes across the IOI. A factor that influences the relationship between $T_{*\delta}$ and $u_{*0}$ is $\Gamma$, since a high value of $\Gamma$ also enables more efficient heat transfer at the IOI. High $\Gamma$ results in rapid cooling of the BL-DSP (Fig. 1g, h) since the vertical mixing of warmer water into the BL is independent of $\Gamma$. For similar magnitudes of $u_{*0}$ and $\Gamma$, due to higher vertical mixing in the HTC scheme, $T_{*\delta}$ in HTC is slightly higher than that in the PP scheme (Fig. 1h) such that the heat fluxes are conserved.

### Calculated vs parameterised melt rates in the IOBCM
We now come to the comparison of melt rates ($\dot{m}$) given by the IOBCM with the melt rates ($\dot{m}_h$) obtained when a BLM parameterisation is used. To emulate the BLM approach in ocean general circulation models, a BL depth h = 30 m (default depth in a previous study[10]) is prescribed and Eqs. 3–5 are used to compute the parameterised melt rate, $\dot{m}_h$, using model output. The melt rate $\dot{m}$ given by IOBCM is as follows:

$$\dot{m} = \left(\frac{c}{L_i}\right)u_{*0}\Gamma T_{*\delta}. \tag{6}$$

Due to the well-mixed nature of the BL-DSP considered, within the BL itself, any depth average values of $T_*$ will be close to, or marginally higher than, $T_{*\delta}$. Hence, for h < $D_{BL}$, the depth averaged $T_*$ ($T_{*h}$) will be a reasonable approximation of $T_{*\delta}$. However, for cases where h > $D_{BL}$, $T_{*h}$ will be higher than $T_{*\delta}$, because of the sharp rise in $T_*$ just beyond $D_{BL}$ (Fig. 1, a, b, c, d) toward its far-field value of $T_{*a}$. Hence, $T_{*h}$ calculated for $D_{BL} <$ h will have artefacts from the ambient ocean leading to considerable differences between $T_{*\delta}$ and $T_{*h}$.

As the BL depth grows with $U_{FF}$, $T_{*\delta}$ as well as the $T_*$ within the BL drops towards 0 (Fig. 1d). Conversely, for lower $U_{FF}$, $T_{*\delta}$ is closer to $T_{a*}$, but then $D_{BL} \ll$ h, and that can give $T_{*h} \gg T_{*\delta}$ leading to $\dot{m}_h \gg \dot{m}$. Those

## Table 1 | Details of datasets generated using the IOBCM

| | Dataset-1 | Dataset-2 |
|---|---|---|
| Purpose | Studying boundary layer dynamics in the HTC and PP mixing schemes | For parameterisation framework using the HTC mixing scheme |
| Total no. of simulations | 480 (240 for each mixing scheme) | 3544 |
| $T_{*a}$ (°C) | 1 and 2 | 0.005, 0.01, 0.02, 0.03, 0.04, 0.05, 0.07, 0.10, 0.15, 0.20, 0.25, 0.50, 1.0, 1.5, 2.0, 2.5, and 4.0 |
| $\Gamma$ | 0.003, 0.006, 0.009, 0.012, and 0.1 | 0.001, 0.003, 0.006, 0.009, 0.012, and 0.1 |
| $\sqrt{C_d}$ | 0.05 and 0.12 | 0.05, 0.055, 0.079 and 0.12 |
| $U_{FF}$(m s$^{-1}$) | 0.0014 to 1.05 | $6.7541*10^{-06}$ to 1.48 |
| $f$(rads s$^{-1}$) | $-1.408*10^{-04}$ | $-1.452*10^{-04}$, $-1.408*10^{-04}$, $-1.321*10^{-04}$, and $-1.194*10^{-04}$ |

principles are illustrated with a series of plots in Fig. 2. In the subplots, it can be observed that while $\dot{m}$ increases gradually with $U_{FF}$, there is a considerable overprediction in values of $\dot{m}_h$ for lower $U_{FF}$. When $U_{FF}$ is close to 0, $T_{*\delta} \cong T_{*a}$ (Fig. 1g), so there is little difference between $T_{*\delta}$ and $T_{*h}$ and consequently between $\dot{m}_h$ and $\dot{m}$. However, as $T_{*\delta}$ drops rapidly with increasing $U_{FF}$, while $T_{*h}$ can remain closer to $T_{*a}$ when $D_{BL}$ remains smaller than h, there is an overestimation of $\dot{m}_h$ for moderate values of $U_{FF}$. Due to the inverse relationship between $\Gamma$ and $T_{*\delta}$, $\dot{m}$ computed by the product of $\Gamma$ and $T_{*\delta}$ shows only weak dependence on $\Gamma$. Since $T_{*h}$ is mainly governed by the difference between $D_{BL}$ and h, the overprediction in $\dot{m}_h$ computed from the product of $\Gamma$ and $T_{*h}$ shows a high dependence on $\Gamma$. This leads to substantial overprediction of melting by $\dot{m}_h$ for high $\Gamma$. This can be observed when rows of Fig. 2 are compared against each other.

In like-for-like comparisons between the HTC and PP schemes, overprediction of $\dot{m}_h$ is greater in PP. This is because, the growth of $D_{BL}$ with $U_{FF}$ is more sluggish in PP. Hence, the differences between $D_{BL}$ and h that determines the difference between $T_{*h}$ and $T_{*\delta}$ are higher in the PP scheme. This leads to higher $\dot{m}_h$ in PP for similar values of $U_{FF}$. The high viscosity in the HTC scheme leads to higher $u_{*0}$ and $D_{BL}$ for similar values of $U_{FF}$ (Fig. 1e, f), so for the range of $U_{FF}$ considered, $D_{BL}$ grows quickly to depths that exceed the prescribed h value of 30 m. However, the maximum value of $D_{BL}$ for PP in the given range of $U_{FF}$ is close to 30 m (Fig. 1e). Hence, to close the gap between $\dot{m}_h$ and $\dot{m}$ in the PP scheme, the IOBCM would need to be forced with higher $U_{FF}$.

Given the above results, the reported sensitivity of melt rates in ocean models to the choice of BL depth[10] is not surprising and is also dependent on mixing schemes used. The best way to minimise overprediction of $\dot{m}_h$ in a BL-DSP is by reducing the size of the prescribed BL depth, h. Experiments could be performed to identify conditions for $D_{BL} < h$. Once such conditions are identified, $\dot{m}_h$ calculated where $D_{BL} = h$ can be set as an upper melt rate cutoff and applied to all cases where $D_{BL} < h$ for slower currents. This would greatly reduce any overestimation of $\dot{m}_h$ for shallow BL-DSPs.

### A parameterisation framework

In a BL-DSP, $T_{*\delta}$ drops as $D_{BL}$ grows due to an increasing $u_{*0}$ (Figs. 1g, h, 3a), so that the lowest attainable $T_{*\delta}$ is physically constrained to a limiting value close to 0 for high $U_{FF}$. Hence, once an expression for the limiting value of $T_{*\delta}$ is obtained, the melt rates, at least for higher $U_{FF}$ can be parameterised just based on $u_{*0}$ (Eq. 6). This opens an avenue for melt parameterisation in BL-DSPs.

Thermal driving within the BL is governed by cooling at the IOI due to the heat lost to melting ice and warming due to mixing of ambient waters into the BL. Considering the nearly uniform $T_*$ inside a well-mixed BL-DSP, the depth averaged $T_*$ inside the BL can be approximated as $T_{*\delta}$. The cooling is quantified as $u_{*0}\Gamma T_{*\delta}$ (Eq. 24 in Methods). At the bottom of the BL (z = $-D_{BL}$), warming depends on the local diffusivity (K) and the local thermal driving gradient $\left(\frac{\partial T_*}{\partial z}\right)$. Because of the sharp nature of the pycnocline, $\frac{\partial T_*}{\partial z}$ can be approximated in a bulk form as $\frac{(T_{*a} - T_{*\delta})}{d_{pyc}}$ where $d_{pyc}$ is the

depth range of the pycnocline. In other words, $d_{pyc} = D_{BL} - D_a$ where $D_a$ is the depth at which the profile of $T_*$ assumes a value close to $T_{*a}$. Similarly, a bulk diffusivity $K_{pyc}$ can be conceptualised for the pycnocline. Based on Eq. 19 (Methods), the rate of change of $T_{*\delta}$ inside the BL can be expressed by the following equation:

$$D_{BL}\frac{dT_{*\delta}}{dt} = u_{*0}\Gamma\left(0 - T_{*\delta}\right) + \frac{K_{pyc}\left(T_{*a} - T_{*\delta}\right)}{d_{pyc}}. \quad (7)$$

In Eq. 7, the first term on the RHS represents the cooling by melting and the second term denotes the warming by vertical mixing. For a BL-DSP in steady state, $\frac{dT_{*\delta}}{dt} = 0$ and with this assumption, an equation for $T_{*\delta}$ can be derived:

$$\frac{T_{*\delta}}{T_{*a}} = \frac{K_{pyc}}{d_{pyc}u_{*0}\Gamma + K_{pyc}}. \quad (8)$$

Surprisingly, $D_{BL}$, an important parameter in most parameterisation is cancelled out in Eq. 8. The efficacy of Eq. 8 has been tested using a large dataset generated by running the IOBCM with the HTC mixing scheme for a wide range of parameters (see dataset-2 in Table 1). The $\frac{T_{*\delta}}{T_{*a}}$ vs log $(u_{*0})$ curve drawn from the dataset is shown in Fig. 3a. Despite the wide range of $T_{*a}$ used, the response of $\frac{T_{*\delta}}{T_{*a}}$ to log $(u_{*0})$ is largely independent of $T_{*a}$. This is promising from a parameterisation point of view.

Since, K is governed by velocity gradients, $K_{pyc}$ can be assumed to be proportional to $u_{*0}$. Hence a constant of proportionality $m_{Kb}$ is introduced in a parametric equation for $K_{pyc}$:

$$K_{pyc} = k_b + m_{kb}u_{*0} \quad (9)$$

The default global K value in IOBCM is $K_b$ $1 \times 10^{-5}$ m$^2$ s$^{-1}$ and it increases with the velocity gradient. Hence, eq. 9 has been formulated to give the required proportionality while ensuring that the minimum value for $K_{pyc}$ is $K_b$. The profiles for $\frac{T_{*\delta}}{T_{*a}}$ given by Eq. 8 (black lines in Fig. 3a) closely trace the true profiles given by IOBCM (coloured dots in Fig. 3a) when $m_{Kb}$ is set to 0.0007. $d_{pyc}$ is set to 8.0 m because for all cases, it is found to rise quicky with rising $u_{*0}$ before plateauing at 8.0 m. The nonlinear reduction in $\frac{T_{*\delta}}{T_{*a}}$ is quicker for higher values of $\Gamma$ as it enables efficient cooling even for lower $T_{*\delta}$ to balance the warming that is primarily driven by $u_{*0}$, thus enabling quick development of a BL-DSP. The diffusivity at the IOI expressed as $K_0 = u_{*0}\Gamma\delta$ (Eq. 22), almost uniquely controls the BL thermal driving (Fig. 3b) owing to its linear relationship with $u_{*0}$. Substituting $K_0$ in Eq. 8 leads to the following relationship:

$$\frac{T_{*\delta}}{T_{*a}} = \frac{\delta K_{pyc}}{d_{pyc}K_0 + \delta K_{pyc}}. \quad (10)$$

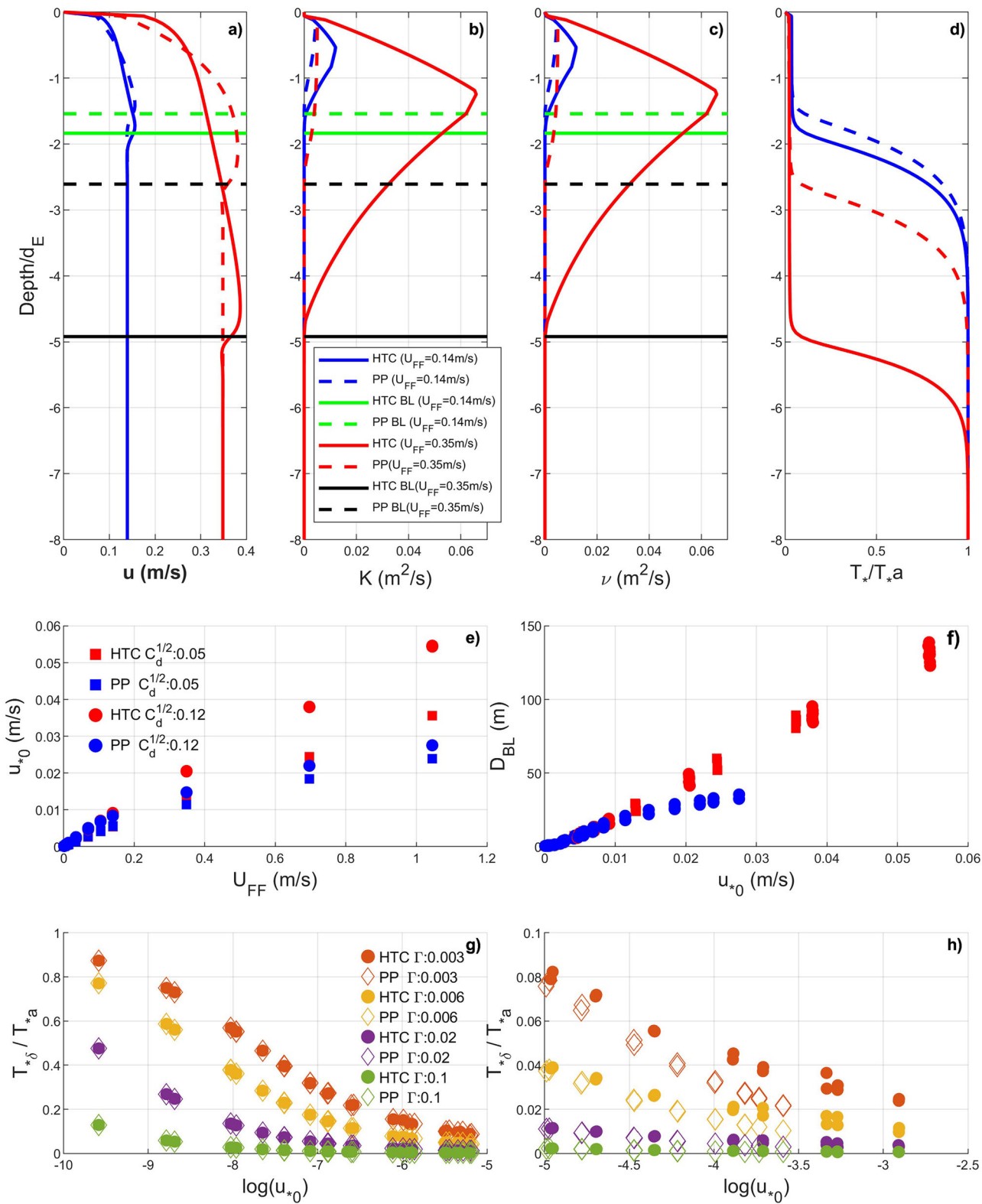

**Fig. 1 | Boundary layer dynamics.** Profiles of (**a**) current speed, (**b**) eddy diffusivity (K), (**c**) eddy viscosity and **d**) thermal driving ratio of $T_*/T_{*a}$ for a low (blue) and a high (red) far field velocity ($U_{FF}$). Broken lines represent the PP mixing scheme and solid lines the HTC mixing scheme. The horizontal lines (black and green) in (**a**), (**b**) and (**c**) represent boundary layer depth ($D_{BL}$). **e** Friction velocity at the IOI ($u_{*0}$) vs far field velocity ($U_{FF}$) for both schemes and two different drag coefficients ($\sqrt{C_d}$ of

0.05 and 0.12) where high $u_{*0}$ corresponds to high $\sqrt{C_d}$. **f** Boundary layer depth ($D_{BL}$) vs friction velocity at the IOI ($u_{*0}$). **g, h** Thermal driving ratio ($T_{*\delta}/T_{*a}$) at a distance $\delta < 0.5$ m from the IOI vs $\log(u_{*0})$. The depths presented in (**a**–**d**) are normalised by Ekman depth $d_E = \sqrt{\frac{2\vartheta_b}{|f|}} = 8.43$ m, where $\vartheta_b$ is the background eddy viscosity.

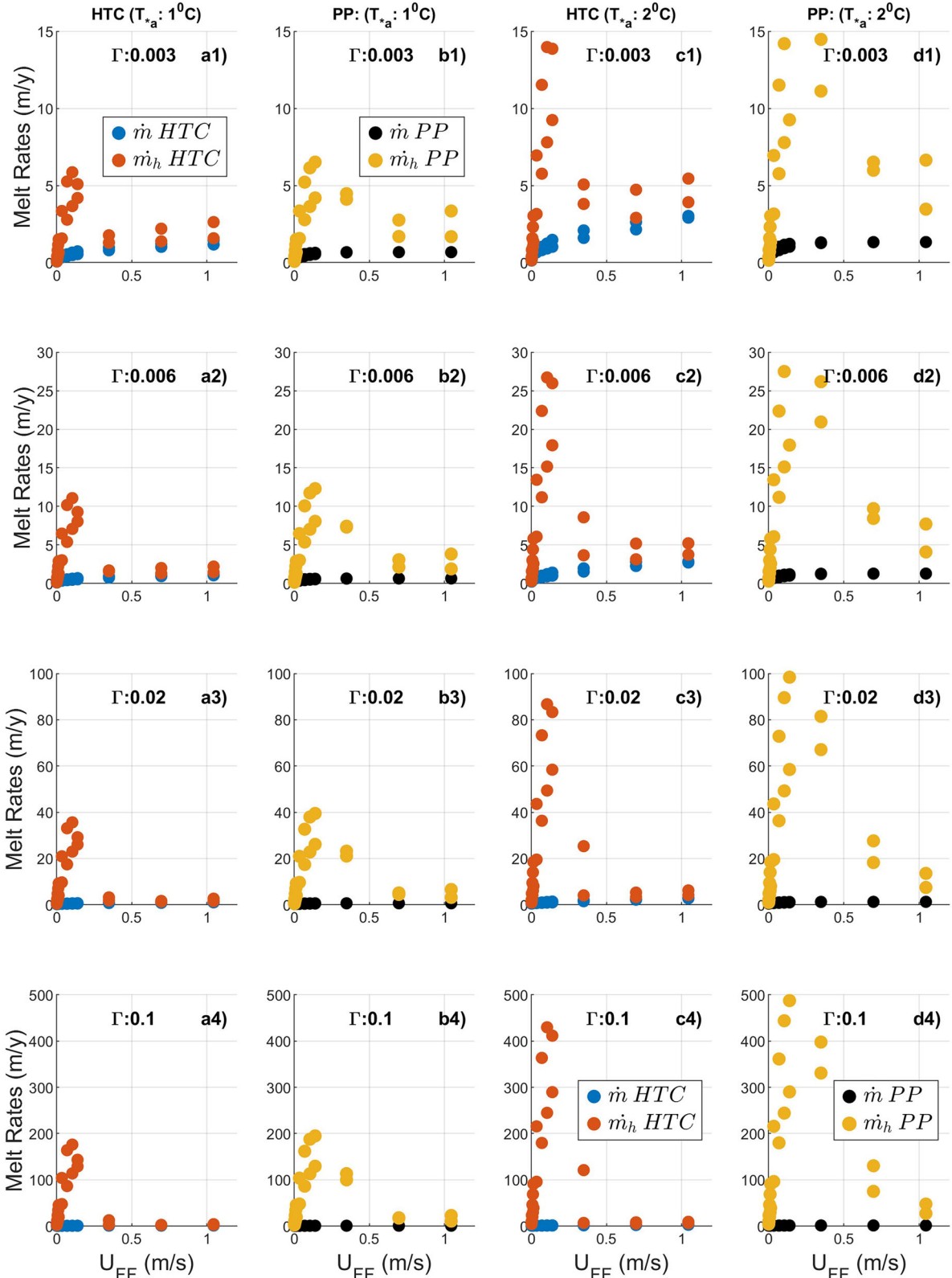

**Fig. 2 | Comparison of calculated($\dot{m}$) and parameterised ($\dot{m}_h$, BLM approach) melt rates in the HTC ($\dot{m}$ in blue and $\dot{m}_h$ in red) and PP ($\dot{m}$ in black and $\dot{m}_h$ in yellow) schemes for different turbulent transfer coefficients (Γ) shown inside plots. a1-4** HTC scheme for an ambient thermal driving ($T_{*a}$) of $1^0$C. **b1-4** PP scheme for an ambient thermal driving ($T_{*a}$) of $1^0$C. **c1-4** HTC scheme for an ambient thermal driving ($T_{*a}$) of $2^0$C. **d1-4** PP scheme for an ambient thermal driving ($T_{*a}$) of 2 °C.

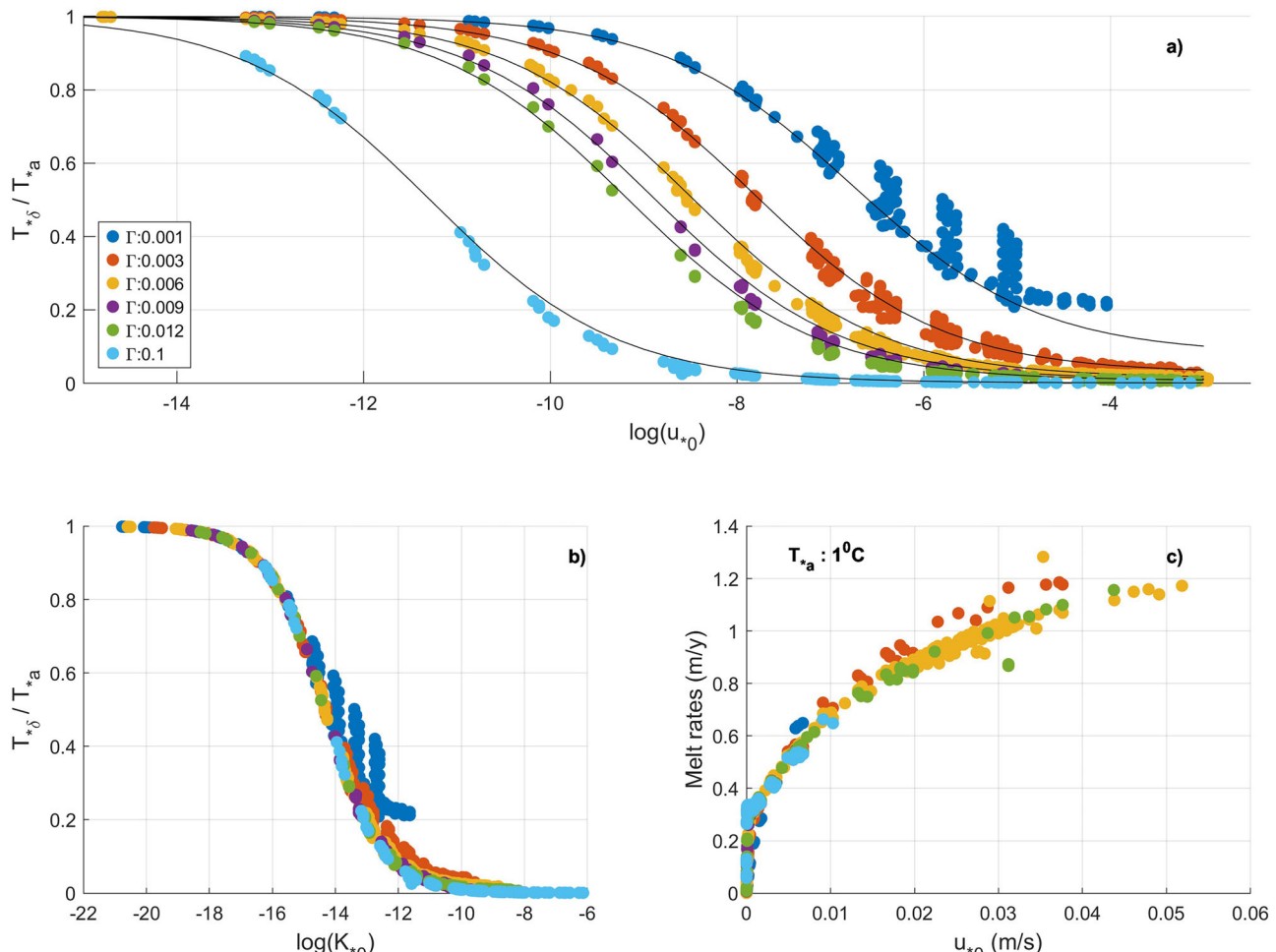

**Fig. 3 | Response of effective thermal driving and basal melting to far field currents. a, b** Respective response of effective thermal driving to friction velocity ($u_{*0}$) and to thermal diffusivity ($K_0$) at the IOI for a range of $T_{*a}$ (0.005 to 4) and $\sqrt{C_d}$ (0.05 to 0.12). **c** Melt rates given by the IOBCM against $u_{*0}$ for $T_{*a} = 1$ °C.

In the above equation, apart from the effective thermal diffusivity at the IOI (Eq. 22), $K_0$, the remaining terms are insensitive to $\Gamma$. Hence $\frac{T_{*\delta}}{T_{*a}}$ takes similar values for all $\Gamma$ for a given value of $K_0$ (Fig. 3b). Thus, the efficiency of heat transfer at the IOI governs the BL thermal driving for BL-DSPs as long as it covaries with vertical mixing, underscoring the BL dynamics discussed earlier. In Fig. 3a, b, there are some deviations from the profiles given by Eq. 8 and eq. 10 respectively. These are cases where values of $T_{*a}$ and $\Gamma$ are extremely low, such that ($T_{*a} - T_{*\delta}$) in Eq. 7 is not an acceptable representation of the $T_*$ gradient within the pycnocline.

Higher $U_{FF}$ or $u_{*0}$ leads to the condition $d_{pyc}u_{*0}\Gamma \gg K_{pyc}$. Ignoring $K_{pyc}$ in the denominator of Eq. 8 yields:

$$T_{*\delta} \cong \frac{K_{pyc}}{d_{pyc}u_{*0}\Gamma} T_{*a}. \tag{11}$$

Substituting this in Eq. 6 (Methods), approximates $\dot{m}$ as:

$$\dot{m} \cong \left(\frac{c}{L_i}\right) \frac{K_{pyc}}{d_{pyc}} T_{*a}. \tag{12}$$

Equation 12 suggests that $\dot{m}$ is independent of $\Gamma$. Figure 3c shows a plot of $\dot{m}$ vs $u_{*0}$ for different $\Gamma$ and underscores the weak dependence of $\dot{m}$ on $\Gamma$ for a horizontal IOI at least in the IOBCM. This could be why ocean model studies[9,10] on melt rate sensitivity to $\Gamma$ are inconclusive, and why the tuning of $\Gamma$ to match basal $\dot{m}$ for BL-DSPs may not always yield favourable outcomes. However, when $\frac{K_{pyc}}{d_{pyc}}$ is comparable to $\Gamma$, $\Gamma$ becomes more relevant, as, for

example, when instabilities associated with sloped IOIs enhance vertical mixing.

Since, $T_{*\delta}$ can be derived from $u_{*0}$ (Eqs. 9 and 10), a parameterisation framework for $\dot{m}$ can be built on parameterisation of $u_{*0}$. A simple approximation for $u_{*0}$, $u_{*FF}$, can be expressed as follows:

$$u_{*FF} = \frac{1}{2}|U_{FF}|\sqrt{C_d}, \tag{13}$$

Note that the above parameterisation for $u_{*FF}$ is case specific and based on a quasilinear relationship (Figs. 1e, 4a). While the constant of proportionality of 1/2 is best suited for the HTC scheme (at least for the range of $\sqrt{C_d}$ values used, between 0.05 and 0.12) it reduces further for the PP mixing scheme (Fig. 1e). The parameterisation of friction velocity is thus mixing scheme dependent. Since $u_{*FF}$ is used in further discussion, $K_{pyc\_FF}$ is introduced, being the same as $K_{pyc}$ but a function of $u_{*FF}$ instead of $u_{*0}$:

$$K_{pyc\_FF} = K_b + m_{Kb}u_{*FF}, \tag{14}$$

$$\dot{m} = \left(\frac{c}{L_i}\right)\left[\frac{K_{pyc\_FF}}{d_{pyc}u_{*FF}\Gamma + K_{pyc\_FF}}\right]u_{*FF}\Gamma T_{*a}. \tag{15}$$

Predicted values of $T_{*\delta}$ are compared with the actual values (y-axis) in Fig. 4b. When $u_{*0}$ is used as a predictor, the values agree well and fall mostly

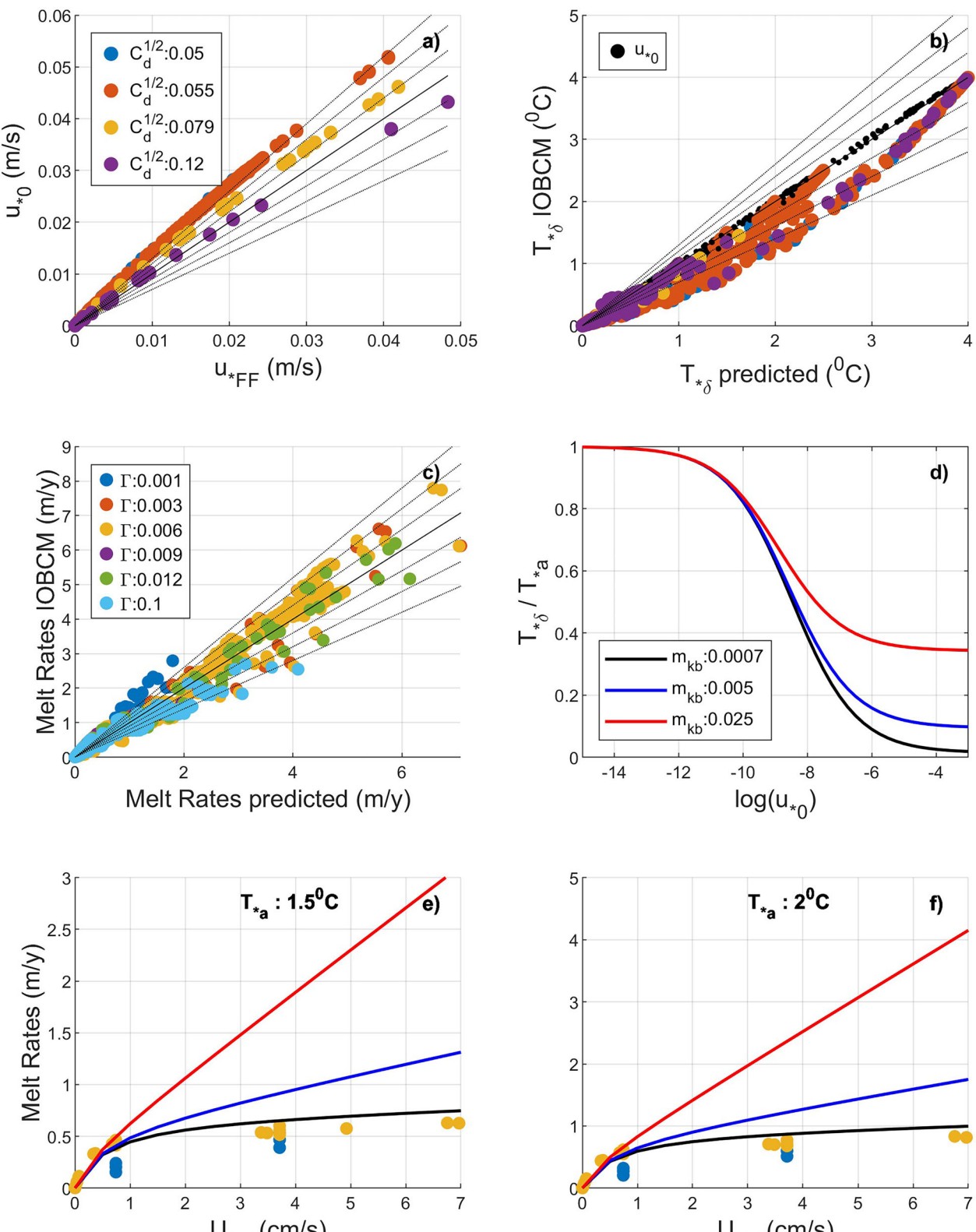

**Fig. 4 | Performance and applicability of the parameterisation framework.**
**a** Comparison between friction velocity at the IOI given by the IOBCM ($u_{*0}$) and
parameterised friction velocity ($u_{*FF}$) **b** Comparison between $T_{*\delta}$ in the IOBCM and
$T_{*\delta}$ predicted when the predictors are $u_{*FF}$ (coloured markers; legends in (**a**) applies
to (**b**) as well) and $u_{*0}$ (black). **c** Comparison of melt rates given by the IOBCM and

predicted melt rates using $u_{*FF}$ as predictor for $T_{*\delta}$. The dotted lines in (**a**), (**b**) and
(**c**) mark 10% increments of the deviation from the zero-error line (solid black).
**d** $T_{*\delta}/T_{*a}$ vs $u_{*0}$ for different values of $m_{Kb}$. **e**, **f** Parameterised $\dot{m}$ for different values
of $m_{Kb}$ and calculated $\dot{m}$ vs $U_{FF}$ for an ambient thermal driving of 1.5 ⁰C and 2.0 ⁰C
respectively. Legends in (**c**) and (**d**) apply to (**e**) and (**f**) as well.

along the zero-error line (black dots). When, $u_{*FF}$ is used as a predictor, for lower values of $u_{*FF}$, where $T_{*\delta} \approx T_{*a}$, the predicted values fall on the zero-error line (coloured dots in Fig. 4b, most noticeable for the case where $T_{*a} = 4$ with $\sqrt{C_d} = 0.005$ $and$ $\sqrt{C_d} = 0.12$). With an increase in $u_{*FF}$, $T_{*\delta}$ deviates further away from the zero-error line before eventually falling back on the line when the limiting lower value is approached. Interestingly, when $u_{*FF}$ is overpredicted $T_{*\delta}$ is underpredicted. Conversely, when $u_{*FF}$ is underpredicted (a small sample in Fig. 4a for higher values of $u_{*FF}$ for $\sqrt{C_d} = 0.12$) $T_{*\delta}$ is over predicted (a small sample in Fig. 4b corresponding to $\sqrt{C_d} = 0.12$). Considering that melt rates are the product of $u_{*FF}$ and $T_{*\delta}$, the inverse relationship between these variables discussed above results in cancellation of error in one variable by the error in the other, at least partially. This could be the reason why melt rate predictions using Eq. 15 are not severely affected by this error except for extreme values of $\Gamma$ (Fig. 4c). Nonetheless, once $T_{*\delta}$ is close to its limiting value (which happens quickly), it remains more or less the same despite changes in $u_{*FF}$. Hence, if $m_{Kb}$ is obtained by fitting predicted melt rates to data, errors will be confined to a regime of weaker currents, or lower $u_{*0}$. Since melting is also low for low $u_{*0}$, satisfactory prediction can be expected. Since, $m_{Kb}$ governs warming by mixing across the pycnocline, it only affects the limiting value of $T_{*\delta}$, as illustrated by the sensitivity of $T_{*\delta}/T_{*a}$ to $m_{Kb}$ for $\Gamma = 0.006$ (Fig. 4d). Hence, just by tuning $m_{Kb}$, parametric models can be fitted to numerical models or observations for this special case of BL-DSP. We limit the scope of this manuscript to this special case. However, for more general geometries, instabilities associated with sloped IOIs enhance vertical mixing which causes an increase in both $m_{Kb}$ and $d_{pyc}$ with slope. Hence, both these parameters would have to be represented as a function of slope to account for slope dependant instabilities/mixing.

## Comparison with observation and other parameterisations

We initially draw comparisons between the IOBCM and the Thwaites grounding zone observations[6], which motivated the study. The $\dot{m}$ versus $U_{FF}$ for cases where $T_{*a} = 1.5$ and $T_{*a} = 2.0$ (which span the observed range 1.66–1.94 °C) are presented in Fig. 4e and 4f respectively. Currents ranging from 2.0 to 5.4 cm s$^{-1}$ were observed under the ice shelf and this yields melt rates between 14.0 and 30.0 m y$^{-1}$ using the canonical model for basal melting[6] versus observed rates of 2.0 to 5.0 m y$^{-1}$. The suppressed basal melting has been ascribed to stratification and weaker currents[6]. While observed melt rates are between 2.0 and 5.0 m y$^{-1}$, melt rates given by both the IOBCM (Fig. 4e, f: dots) and the parametrised IOBCM (Fig. 4e, f: black lines) are conservative with a maximum of 1 m y$^{-1}$. However, if $m_{Kb}$ is increased to 0.025 in the parameterised model, $\dot{m}$ can be increased to the ballpark of observed melting (red line), tuning that might account for the ice shelf slope. For a general parameterisation using Eq. 15, instead of using constant values for $m_{Kb}$ and $d_{pyc}$, they should be expressed as a function of IOI slope. The basic case for flat ice shelves is presented to show the versatility of this parameterisation framework for the case of a BL-DSP.

We next draw comparisons with a recent study[15] (hereafter Y24), which followed an alternative approach to the parameterisation of basal melting for cases when mixing is suppressed by stratification, adopting a functional form for $\Gamma$ based on Large Eddy Simulations. The reduced form of the parameterised $\Gamma_T$ and $\Gamma_S$ (for T and S respectively) in recent studies[15,16], $\Gamma_{para}$ can be expressed as:

$$\Gamma_{para} = \frac{K \frac{\partial T_*}{\partial z}}{u_{*0} T_{*a}}. \tag{16}$$

The variation of $\Gamma_{para}$ with respect to $L^+$ (the ratio of Obukhov to viscous length scales, which is proportional to $u_{*0}{}^4$) is similar but inverse in nature (see Fig. 1 of Y24) to the variation of $T_{*\delta}$ with respect to $\log(u_{*0})$ shown in Fig. 3a. The variation of $\Gamma_{para}$ and $T_{*\delta}$ are only observed for lower $U_{FF}$ and they plateau thereafter. Hence, in Y24, $\Gamma_{para}$ is parameterised as an increasing function of $L^+$ until it reaches a cutoff value $\Gamma_{para\_CC}$ and remains constant thereafter. The cutoff/threshold value ($\Gamma_{T,CC}$ $and$ $\Gamma_{S,CC}$ in Y24) is presented as an upper limit for the stratification feedback. A similar

approach in the current study would result in fixing a lower cutoff value for $T_{*\delta}$ after a certain value for $u_{*0}$ and fitting an equation for its reduction in the low $u_{*0}$ or $U_{FF}$ regime. However, by conserving heat fluxes into the BL in Eq. 7 and focussing on $T_{*\delta}$, this variation and the associated physics are naturally preserved in Eq. 15 without introducing cutoffs/thresholds.

Figure 5 shows a comparison of the parameterisations of Y24 and Eq. 15 against observations (Table B1 of Y24). With $m_{Kb} = 0.025$, the $\dot{m}$ in Eq. 15 is lower than Y24 and the observed melt rates. For $U_{FF} \geq 0.1$ m s$^{-1}$ Y24 overestimates $\dot{m}$ while Eq. 15 underpredicts the same. However, a very high $m_{Kb}$ of 10 in Eq. 15 yields $\dot{m}$ that matches well with observation. Such a high value for $m_{Kb}$ implies that, in all cases, $T_{*\delta}$ is close to $T_{*a}$, and that any further increase in $m_{Kb}$ has little to no effect on $T_{*\delta}$ and $\dot{m}$, as the maximum possible value for $T_{*\delta}$ is $T_{*a}$. Hence, in this case we suspect that the $T_{*a}$ sampled by the borehole observations could be closer to values inside the BL than to those in the far-field ocean. Such a high $m_{Kb}$ does not overestimate $\dot{m}$ for low $U_{FF}$ because the equations constrain the value of $T_{*\delta}$ between 0 and $T_{*a}$.

Clearly, both the parameterisations of Y24 and Eq. 15 have some merit. The former is built on a much more complete physical model, while the simplicity of the latter allows for flexible tuning. They are, however, quite different in concept. Y24, in common with other recent studies, focuses on improving the melt rate computed from Eq. 6 through a consideration of the boundary layer physics subsumed into the turbulent transfer coefficient, $\Gamma_{para}$. In contrast, Eq. 15 exploits the principle that for a horizontal IOI, lateral supply of heat must be negligible inside the BL, so $T_{*\delta}$ will adjust until vertical heat fluxes are balanced. Thus, unlike $\Gamma_{para}$, Eq. 15 allows a physics-based control on the melt rate computation, exploiting the principal that, for a flat IOI, the heat flux through the pycnocline becomes the controlling factor, since diffusivity beyond the boundary layer will always be lower than within it. While Eq. 15 remains simple, it offers a two-way control on the melt rate, one through the IOI by means of $\Gamma$ and the other through the pycnocline by means of $m_{Kb}$ and $d_{pyc}$. Conventional parameterisations lack this control since these fluxes are not explicitly distinguished in their derivations. Also, the influence of $\Gamma$ inferred from Fig. 3a could provide insight into the actual values of $\Gamma$ once more observations are available.

## Potential for advanced framework and process studies

The stratification under a low IOI slope considered here implies quasi-horizontal isopycnals and slow flow, implying negligible lateral heat advection within the BL. Below the pycnocline, vertical mixing is minimal and heat loss to the BL through the pycnocline should therefore be supplied through lateral heat advection below the pycnocline controlled by the large-scale sub-ice circulation. Hence, while lateral advection drives melting, the pathway to the IOI is through the pycnocline. Observations[17] from depths below the pycnocline validate this. Conveniently, in the IOBCM, $T_{*a}$ which in the real world is maintained by lateral fluxes, is maintained by fixing its value. Hence, there is an implicit assumption that lateral advection below the pycnocline is the ultimate source of heat for melting, but that melting is regulated by diffusion through the pycnocline. That assumption necessitates the redirection of primary focus from the BL and the IOI to mixing across the pycnocline through $m_{Kb}$ in Eq. 15. Similarly, a recent numerical study[18] of ice-ocean BL mixing reported that while fluxes across the IOI could be captured with relatively coarse near-interface vertical grid resolution, finer resolutions in the regions of entrainment were critical.

While our parameterisation is based on a constant $T_{*a}$, more complicated settings, such as when dense waters from inside rifts sink and cool down the ambient waters[19], can also be accounted for using Eq. 15 by introducing functions adjusting $T_{*a}$. The versatility of the parameterisation framework also allows inclusion of other contributions to the heat flux across the IOI. For example, heat conduction into the ice shelf[20] can be simply incorporated (Methods Eqs. 25 and 26). While the inclusion of tides in ocean models[14] increases the melt rate through a dynamical component related to increased friction velocity, a third of that increase is negated by a thermodynamic component (see methods, Eqs. 27–30) associated with cooling of the BL. Although the tendency of the BL to cool with increasing

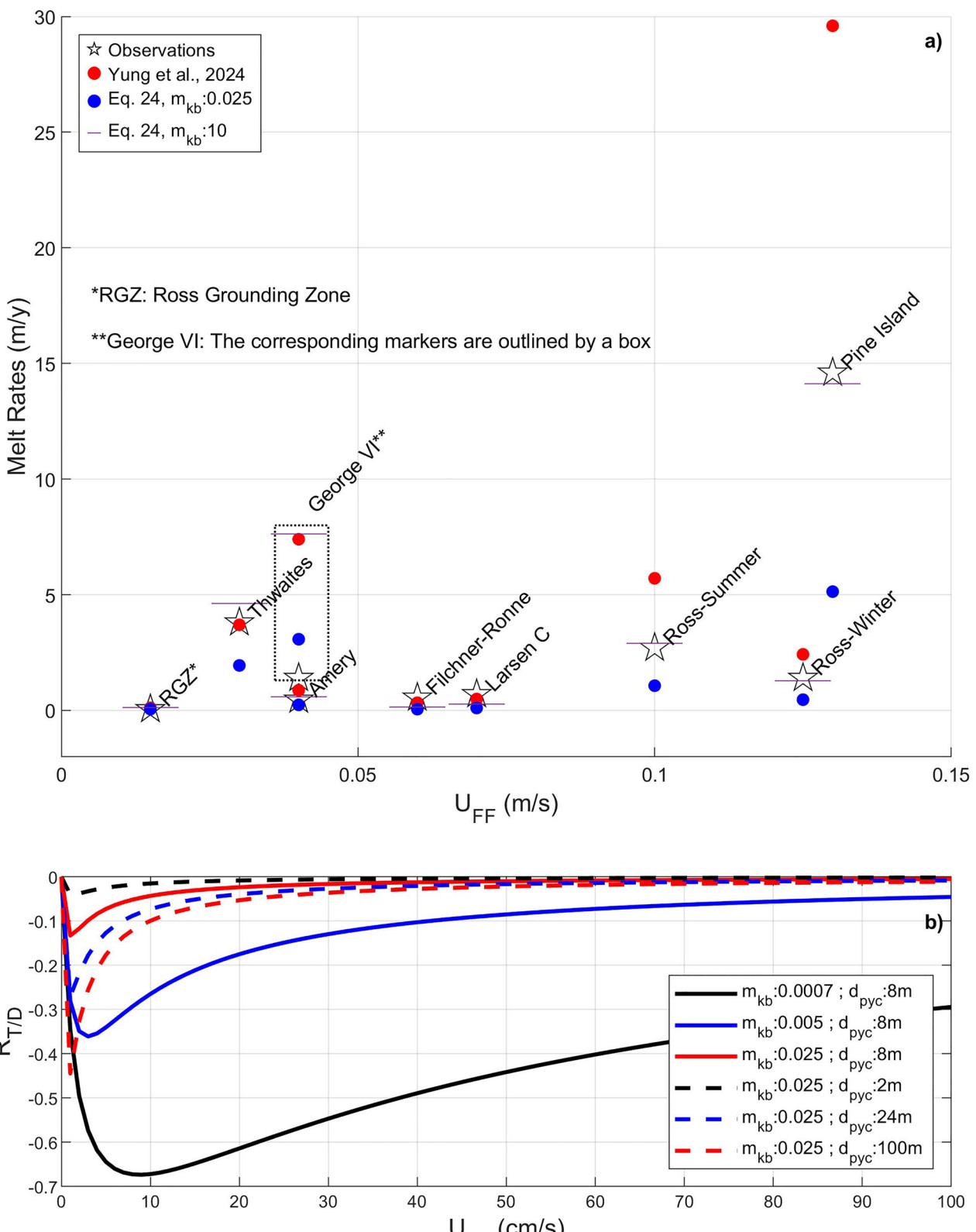

**Fig. 5 | Performance of the parameterisation framework. a** Comparison between borehole observations, basal melt parameterisation from Y24 and current parameterisation framework for different far field velocities ($U_{FF}$). The data are based on melt rates, $U_{FF}$ and far field ambient thermal driving compiled by Y24. **b** $R_{T/D}$ (ratio of thermodynamical component of melt rate change to dynamical component, see Methods, Eq. 29) vs $U_{FF}$ for different $m_{Kb}$ and $d_{pyc}$.

friction velocity is implicitly accounted for, the effect has often been overlooked, leading to the conclusion that non-local processes cause the BL cooling[14]. Here we investigate the ratio of the thermodynamical to dynamical components ($R_{T/D}$) of the melt rate change with changes in friction velocity within our parameterisation framework (Fig. 5b; the relevant parameters are adopted from Jourdain et al.[14]). It is possible for the thermodynamical component associated with the local mixing processes considered in our parameterised IOBCM to negate over half of the dynamical component, so they could certainly account for the third already reported[14]. However, the impact of the thermodynamical component reduces for higher values of $m_{Kb}$, and hence with higher values of IOI slope, since increased slope mixing enhances mixing across the pycnocline.

Observations under Thwaites glacier show that melting increases with the slope of the IOI[7]. In principle, such geometrical control of melting could be accounted for by expressing $m_{Kb}$ as a function of slope based on observations and/or models more complex than IOBCM. However, such a development is beyond the scope of the present study, where we seek to present a parameterisation framework and demonstrate its versatility.

## Conclusions

We have investigated the current basal melt parameterisation approach in z-coordinate ocean models for a stratified water column yielding a boundary layer (BL) over a dynamically stable pycnocline. The work was motivated by a recent study[6] that reported that the conventional model of ice shelf basal melting overestimates the observed basal melting (2.0–5.0 m y$^{-1}$) under eastern Thwaites glacier (as high as 14.0 to 30.0 m y$^{-1}$). It has been reasoned that weak mixing due to low current speed and strong density stratification under the ice-ocean interface (IOI) suppress the heat transport from the warm ambient ocean below the pycnocline. Hence, we revisited the current basal melt parameterisation in ocean models for the simple case of a horizontal IOI, yielding a BL over a dynamically stable pycnocline (BL-DSP).

We use an ice-ocean boundary current model (IOBCM) that parameterises turbulent processes using a mixing length approach. For horizontal or nearly flat IOIs there is no escape route for the buoyant colder melt waters to flow away from their point of origin. This results in a BL of fresher, buoyant and colder waters below the IOI. Therefore, the effective thermal driving ($T_{*\delta}$) in the vicinity of the IOI (i.e. at a distance $\delta \cong 0.5m$) is much less than the ambient thermal driving ($T_{*a}$), thus minimizing the basal melting. The IOBCM is first evaluated for its ability to capture this necessary physics using two mixing schemes. The behaviour of the BL under both mixing schemes is similar, with mixing inside the BL spreading cold and buoyant melt waters over the depth of the BL. In both mixing schemes, thermal driving ($T_{*\delta}$) falls and BL depth ($D_{BL}$) grows as the far field current velocity ($U_{FF}$) increases.

We compare melt rates calculated with the IOBCM ($\dot{m}$) against those obtained when a basal melt parameterization approach, as used in z-coordinate ocean models[8,9], is applied to IOBCM output ($\dot{m}_h$). In that approach, a BL depth (h) is prescribed and average ocean properties within that distance of the IOI are used for basal melt calculations. Hence, we refer to this as a boundary layer model (BLM). Through comparison of parameterised melt rates ($\dot{m}_h$) from the BLM and calculated melt rates ($\dot{m}$) from the IOBCM for a BL-DSP, we show that for cases where $D_{BL} < h$, i.e., for lower $U_{FF}$, $\dot{m}_h$ is overpredicted. The over prediction depends on the mixing scheme used, the turbulent transfer coefficient ($\Gamma$) used and ambient ocean properties. While reducing h can minimize this error, it is important to identify conditions/regimes that lead to a $D_{BL} < h$. Applying melt rates observed when $D_{BL}$ is slightly greater than h for all cases where $D_{BL} < h$ could be a quick fix worth trying.

The rapid reduction in $T_{*\delta}$ with increase in far field ocean current velocity ($U_{FF}$) in a BL-DSP leads to $T_{*\delta}$ acquiring a limiting value close to 0. This means, once $T_{*\delta}$ drops to its limiting value, it remains more-or-less the same for higher $U_{FF}$. This makes the parameterisation of ice-ocean heat transfer in a BL-DSP easier. Consequently, we propose a parameterisation framework for ocean-ice heat transfer and melting that addresses this specific case. In this approach, just by assuming the well-mixed BL-DSP is in a steady

state, an expression for the thermal driving inside the BL can be derived by equating heat fluxes across the BL, namely the cooling at the IOI due to heat loss from the BL to ice and the warming due to mixing of warm ambient ocean waters into the BL. Assuming that the mean thermal driving inside the BL is the same as $T_{*\delta}$, we show that a parametric equation for $T_{*\delta}$ and $\dot{m}$ can be derived that retains the necessary physics. In the development of this framework, we also show that the sensitivity of $\dot{m}$ to $\Gamma$ is weak in certain cases.

We also show the versatility of the parameterisation framework by comparing parameterised melt rates with calculated melt rates in the IOBCM and for a range of observations. We show that it is easy to tune the parametric model to observations or models for the specific case of a horizontal IOI. In this framework, warming by mixing across the pycnocline is the most important element, and it is conceptualised by parameterising 1) a bulk diffusivity for the pycnocline below the BL and 2) a thermal driving gradient across the pycnocline.

The versatility of the parameterisation framework lies in the two-way physics-based control on ice-ocean heat exchange in Eq. 15 through fluxes across the IOI and the pycnocline. Control of heat flux across the IOI is achieved by means of the turbulent transfer coefficient, $\Gamma$. The same across the pycnocline is achieved through parameters $m_{Kb}$ (mixing across the pycnocline) and $d_{pyc}$ (pycnocline depth). In this introductory work on the parameterisation framework, the simple case of a flat IOI allows usage of constant values for these parameters. For more complicated settings of BL-DSP, such as when the IOI is sloped, both should be represented as functions of slope. While that requires further analysis of observations and model output, the framework presented here lays a useful foundation. Owing to the simplicity of Eq. 15, modelers can try fitting available data to it and make informed deductions towards improvement of these parameters including $\Gamma$. The underlying framework provided is versatile, allowing for subsequent validation and expansion, which will be explored in follow up work.

## Methods

### Ice-ocean boundary current model

The IOBCM is conceptualized as a 1D model[3,13] for heat transfer from a warm ambient ocean to the IOI in the direction perpendicular (z axis) to the IOI. The spatial variation in the x and y directions that are parallel to the IOI are neglected. While the original IOBCM[3] is a simple 1D model with constant viscosity and diffusivity, a further development[13] includes a turbulence closure that computes eddy viscosities ($\vartheta$) and diffusivities (K) from currents and stratification. The later IOBCM[13] discussed here uses the hybrid turbulence closure (HTC). HTC combines the Local Turbulence Closure[21,22] (LTC), based on Monin-Obukhov similarity theory, applied in the mixed layer, with the classic Pacanowski and Philander (PP) mixing scheme[23], applied in the stratified layer below. For brevity, only the basics of the IOBCM are presented here. Readers are referred to the original manuscript describing IOBCM[13] for full details including representative values of various constants used in the model equations presented below.

$$\frac{\partial u}{\partial t} - fv = -fv_{FF} + \frac{\partial \left(\vartheta \frac{\partial u}{\partial z}\right)}{\partial z} \tag{17}$$

$$\frac{\partial v}{\partial t} + fu = fu_{FF} + \frac{\partial \left(\vartheta \frac{\partial v}{\partial z}\right)}{\partial z} \tag{18}$$

$$\frac{\partial T_*}{\partial t} = \frac{\partial \left(K \frac{\partial T_*}{\partial z}\right)}{\partial z} \tag{19}$$

Boundary conditions are as follows.

*At the IOI where $z = 0$ : $u = 0, v = 0, T_* = 0$*

*In the far field where $z = -\infty$ : $u = u_{FF}, v = v_{FF}, T_* = T_{*a}$*

$$T_{*a} = T_a - (\lambda_1 S_a + \lambda_2 + \lambda_3 Z)$$

The equations above are solved for variables u, v and $T_*$ as functions of time t along the z axis. $T_{*a}$ here represents thermal driving[24] derived from ambient ocean temperature ($T_a$) and salinity ($S_a$). Using the thermal driving as a variable yields a simple boundary condition at the IOI of 0. Velocities in x (upslope) and y (across slope) directions along the IOI are denoted $u(z)$ and $v(z)$ respectively.

In Eqs. 17 and 18, the first terms on the RHS represent the acceleration in the x and y directions respectively due to geostrophic far field currents ($u_{FF}$ and $v_{FF}$ respectively) that are driven by the deviation of the ice shelf from its level of free floatation and deflected by the Coriolis effect. The second terms are viscous resistance. An additional term to account for density driven currents along a sloped IOI in Eq. 17 is omitted here as our study is limited to flat IOIs. For convenience, the forcing on the system is expressed here as $fv_{FF}$ and $fu_{FF}$ where $f$ is the local Coriolis parameter. Those terms are originally[3,13] $g\cos\alpha\frac{\partial\eta}{\partial x}$ and $g\cos\alpha\frac{\partial\eta}{\partial y}$ respectively, where $\eta$ is the vertical deviation of the IOI from its equilibrium position. In Eq. 19, the profile for $T_*$ is determined solely by the eddy diffusivity, K which is dependent on u, v and $T_*$ itself.

The LTC is a mixing length model[13], in which both $\vartheta$ and K are linear functions of mixing length and friction velocity, $u_*$. The mixing length grows with distance from the IOI, reaching a limiting value that is set by stratification and Earth rotation, while $u_*$ is computed as follows:

$$\text{Friction velocity, } u_*(z) = \left(\vartheta\left|\frac{\partial \boldsymbol{u}(z)}{\partial z}\right|\right)^{\frac{1}{2}}. \quad (20)$$

### Ice ocean interface
Based on observations in turbulent boundary layer flows, the effective thermal diffusivity ($K_0$) and viscosity ($\vartheta_0$) in the vicinity of a boundary or wall are expressed as linear functions of $u_*$ at the IOI ($u_{*0}$) as follows[13]:

$$\text{Friction velocity at IOI, } u_{*0} = \left(\vartheta_0\left|\frac{\boldsymbol{u}_\delta}{\delta}\right|\right)^{\frac{1}{2}}, \quad (21)$$

$$K_0 = u_{*0}\Gamma\delta, \quad (22)$$

$$\vartheta_0 = u_{*0}\sqrt{C_d}\delta. \quad (23)$$

The subscripts in the above equations represent model grid nodes, where 0 represents the IOI and $\delta$ the first grid point below, a small distance ($\delta < 0.5$m) away from the IOI. From Eqs. 22 and 19, the heat flux at the IOI can be expressed as:

$$q = u_{*0}\Gamma T_{*\delta}, \quad (24)$$

leading to an expression for melt rate at the IOI given by Eq. 6.

### Heat conduction into the ice
Additionally, along with turbulent heat transport across the IOI, it is possible to include heat conduction into the ice shelf in Eq. 7. Following a simple model[20] for heat conduction into the ice shelf, Eq. 7 can be modified as follows:

$$D_{BL}\frac{dT_{*\delta}}{dt} = u_{*0}\Gamma\left(0 - T_{*\delta}\right) - \frac{\rho_i c_i K_i}{\rho c}\left(\frac{\partial T_i}{\partial z}\right) + \frac{K_{pyc}\left(T_{*a} - T_{*\delta}\right)}{d_{pyc}}, \quad (25)$$

In the above equation $\rho_i$, $c_i$, and $K_i$ refer to density of ice, specific heat capacity of ice and thermal conductivity of ice respectively, $\rho$ denotes density

of sea water and $T_i$ denotes far-field temperature inside the ice shelf. Assuming a steady $T_{*\delta}$, this leads to the following expression for $T_{*\delta}$:

$$T_{*\delta} = \frac{K_{pyc}T_{*a} - d_{pyc}\frac{\rho_i c_i K_i}{\rho c}\left(\frac{\partial T_i}{\partial z}\right)}{d_{pyc}u_{*0}\Gamma + K_{pyc}}. \quad (26)$$

A model that also captures heat conduction into the ice shelf can be trained using the above equation. Following the parameterisation framework in this manuscript, $K_{pyc}$ and $d_{pyc}$ can be parameterised by fitting the model results to the above equation.

### Decomposition of melt rate change
The underlying physics of the parameterised equation allows us to draw informed deductions concerning ocean model experiments[14] for Amundsen Sea ice shelves with and without tidal forcing. The influence of tides on melt rates was studied by decomposing the change in melt rate into dynamical and thermodynamical components[14]. Decomposition of Eq. 6 yields the following equation for the change in melt rate:

$$\Delta\dot{m} = \left(\frac{c}{L_i}\right)\Gamma\Delta u_{*0}T_{*\delta} + \left(\frac{c}{L_i}\right)\Gamma u_{*0}\Delta T_{*\delta} + \left(\frac{c}{L_i}\right)\Gamma\Delta u_{*0}\Delta T_{*\delta}, \quad (27)$$

In the above equation, the first term denotes the dynamical term, where $\Delta u_{*0}$ represents the change in friction velocity due to the inclusion of tides. Similarly, the second term denotes the thermodynamical term where $\Delta T_{*\delta}$ represents the change in BL thermal driving due to the inclusion of tides. The last term is referred to as a covariational component. The above decomposition ignores the strong coupling between $u_{*0}$ and $T_{*\delta}$ due to the growth of the BL. Hence, the relatively large contribution of the thermodynamic and covariational components (nearly one third of the dynamical) were attributed to non-local processes or tide-induced cooling of the continental shelf[14]. A similar decomposition considering the coupling between variables can be done by means of Eqs. 6, 8 and 9. From Eq. 6:

$$\frac{d\dot{m}}{du_{*0}} = \left(\frac{c}{L_i}\right)\Gamma T_{*\delta} + \left(\frac{c}{L_i}\right)u_{*0}\Gamma\frac{dT_{*\delta}}{du_{*0}}, \quad (28)$$

Multiplying the above equation with $\Delta u_{*0}$:

$$\Delta\dot{m} = \left(\frac{c}{L_i}\right)\Gamma T_{*\delta}\Delta u_{*0} + \left(\frac{c}{L_i}\right)u_{*0}\Gamma\frac{dT_{*\delta}}{du_{*0}}\Delta u_{*0}. \quad (29)$$

Since $\frac{dT_{*\delta}}{du_{*0}}\Delta u_{*0} = \Delta T_{*\delta}$, the equation is similar to Eq. 6 wherein the first term can be inferred as the dynamical component and the second as the thermodynamical component. From Eq. 8 and 9 (also resubstituting $T_{*\delta}$ back after differentiation):

$$\frac{dT_{*\delta}}{du_{*0}} = -T_{*\delta}\frac{d_{pyc}k_b\Gamma}{\left(d_{pyc}u_{*0}\Gamma + k_b + m_{kb}u_{*0}\right)\left(k_b + m_{kb}u_{*0}\right)}, \quad (30)$$

From Eqs. 29 and 30, a ratio of thermodynamical to dynamical terms can be expressed as follows:

$$R_{T/D} = -\frac{u_{*0}d_{pyc}K_b\Gamma}{\left(d_{pyc}u_{*0}\Gamma + K_b + m_{Kb}u_{*0}\right)\left(K_b + m_{Kb}u_{*0}\right)}. \quad (31)$$

### Datasets
Datasets are created by running a range of IOBCM simulations with varying model parameters. The parameter space of the datasets consists of ambient thermal driving ($T_{*a}$), turbulent transfer coefficient ($\Gamma$), drag coefficient ($\sqrt{C_d}$), far-field current speed ($U_{FF}$) and Coriolis parameter ($f$). Two

separate datasets (Table 1) are created for this manuscript. The simulations are run until the model assumes a steady state (i.e., up to 90 inertial cycles). The datasets are created by time averaging model variables over the last 10 (81st to 90th) inertial cycles.

## Reporting summary

Further information on research design is available in the Nature Portfolio Reporting Summary linked to this article.

## Data availability

The data used in this study are generated by solving the model detailed in Jenkins (2021) using Matlab. The generated datasets are available at https://doi.org/10.5281/zenodo.14628475.

## Code availability

The model code is available at https://doi.org/10.5281/zenodo.13381662. Codes to reproduce the figures in this study are available at https://doi.org/10.5281/zenodo.14628475. All variables in this manuscript are detailed in Jenkins (2021, https://researchportal.northumbria.ac.uk/en/publications/shear-stability-and-mixing-within-the-ice-shelf-ocean-boundary-cu), with their respective values given in Table 1.

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

## Acknowledgements

This research is a part of the PROTECT (Projecting Sea-Level Rise: from Ice Sheets to Local Implications: protect-slr.eu) project supported by funding from the European Union's Horizon 2020 research and innovation program (grant agreement 869304) and also received support from the UK Natural Environment Research Council (grant NE/L007037/1).

## Author contributions

T.J and A.J. conceived the study. T.J. undertook the model runs, developed the parameterisation and led the writing the paper. A.J. assisted with the interpretation of the data and writing of the paper.

## Competing interests

The authors declare no competing interests.
