## [Transparent Peer Review file · Communications Earth & Environment]

Physics-based Parameterisation framework for Basal Melting in Ice-Ocean Boundary Layers over dynamically stable Pycnoclines

Corresponding Author: Professor Adrian Jenkins

This manuscript has been previously reviewed at another Nature Portfolio journal. This document only contains reviewer comments and rebuttal letters for versions considered at Communications Earth & Environment.

Version 0:

Decision Letter:

Dear Professor Jenkins,

Your manuscript titled "Revisiting Ice shelf Basal Melt Parameterisations for Ice-Ocean Boundary Layers over dynamically stable Pycnoclines" has now been seen by 3 reviewers, and we include their comments at the end of this message. They find your work of interest, but some important points are raised. We are interested in the possibility of publishing your study in Communications Earth & Environment, but would like to consider your responses to these concerns and assess a revised manuscript before we make a final decision on publication.

We therefore invite you to revise and resubmit your manuscript, along with a point-by-point response that takes into account the points raised. Please highlight all changes in the manuscript text file.

To be considered for publication, along with addressing the concerns of the reviewers, we request that:

- The results and implications to be more clearly described in the abstract and conclusion.

Please submit your point-by-point responses as a separate file, distinct from your cover letter where you can add responses to the Editors' comments that you do not want to be made available to the reviewers. Word files are preferred. We recommend that any figures, tables or graphs that are included in the response to reviewers are also included in the main article or Supplementary Information.

Please use the following link to submit your revised manuscript, point-by-point response to the referees' comments (which should be in a separate document to any cover letter), a tracked-changes version of the manuscript (as a PDF file) and the completed checklist:

Link Redacted

We hope to receive your revised paper within six weeks; please let us know if you aren't able to submit it within this time so that we can discuss how best to proceed. If we don't hear from you, and the revision process takes significantly longer, we may close your file. In this event, we will still be happy to reconsider your paper at a later date, as long as nothing similar has been accepted for publication at Communications Earth & Environment or published elsewhere in the meantime.

Please do not hesitate to contact us if you have any questions or would like to discuss these revisions further. We look

forward to seeing the revised manuscript and thank you for the opportunity to review your work.

Best regards,

Jennifer Veitch, PhD
Editorial Board Member
Communications Earth & Environment
orcid.org/0000-0003-2544-1243

Alireza Bahadori, PhD
Associate Editor
Communications Earth & Environment
Consulting Editor
Communications Sustainability

EDITORIAL POLICIES AND FORMATTING

Editorial Policy: [Policy requirements](https://www.nature.com/documents/nr-editorial-policy-checklist.pdf) (Download the link to your computer as a PDF.)

- Behavioural and social science
- Ecological, evolutionary & environmental sciences
- Life sciences

<https://www.nature.com/documents/nr-reporting-summary.zip>

Furthermore, please align your manuscript with our format requirements, which are summarized on the following checklist: [Communications Earth & Environment formatting checklist](https://www.nature.com/documents/commsj-phys-style-formatting-checklist-article.pdf)

and also in our style and formatting guide [Communications Earth & Environment formatting guide](https://www.nature.com/documents/commsj-phys-style-formatting-guide-accept.pdf) .

*** DATA: Communications Earth & Environment endorses the principles of the Enabling FAIR data project (<http://www.copdess.org/enabling-fair-data-project/>). We ask authors to make the data that support their conclusions available in permanent, publically accessible data repositories. (Please contact the editor if you are unable to make your data available).

All Communications Earth & Environment manuscripts must include a section titled "Data Availability" at the end of the Methods section or main text (if no Methods). More information on this policy, is available at <http://www.nature.com/authors/policies/data/data-availability-statements-data-citations.pdf>.

If a community resource is unavailable, data can be submitted to generalist repositories such as [figshare](https://figshare.com/) or [Dryad Digital Repository](http://datadryad.org/). Please provide a unique identifier for the data (for example a DOI or a permanent URL) in the data availability statement, if possible. If the repository does not provide identifiers, we encourage authors to supply the search terms that will return the data. For data that have been obtained from publically available sources, please provide a URL and the specific data product name in the data availability statement. Data with a DOI should be further cited in the methods reference section.

REVIEWER COMMENTS:

Reviewer #1 (Remarks to the Author):

See the attached file.

Reviewer #2 (Remarks to the Author):

See the attached file.

Reviewer #3 (Remarks to the Author):

The paper by Jayasankar and Jenkins describes a simple parameterisation for calculating the rate of basal melting in the ice-ocean boundary layer beneath nearly horizontal ice shelves where the boundary layer is bounded by a sharp pycnocline. Their model builds on previous modelling of the ice-ocean boundary layer and represents a significant advancement in the field of ice-ocean interactions. Their model can represent the competing effects on basal melting of strong density stratification at the ice base and mixing driven by far field flow speeds. Unlike conventional models of basal melting which overpredict the basal melt rate in the presence of strong stratification and slow far-field current speeds, the model of Jayasankar and Jenkins correctly predicts a highly suppressed rate of melting. Their model can recreate observed rates of basal melting beneath the eastern Thwaites glacier grounding zone, demonstrating that their model has wide applicability.

The work presented in this paper will have significant implications for modelling ice-ocean interactions and therefore I recommend it as suitable for publication. There were a few locations in the text and figures, however, where I felt that a little extra clarity or improved presentation was required. Once these areas have been addressed the paper will be ready for publication.

I have highlighted my areas of concern, along with minor corrections that are required, in the attached pdf.

Communications Earth & Environment is committed to improving transparency in authorship. As part of our efforts in this direction, we are now requesting that all authors identified as 'corresponding author' create and link their Open Researcher and Contributor Identifier (ORCID) with their account on the Manuscript Tracking System prior to acceptance. ORCID helps the scientific community achieve unambiguous attribution of all scholarly contributions. You can create and link your ORCID from the home page of the Manuscript Tracking System by clicking on 'Modify my Springer Nature account' and following the instructions in the link below. Please also inform all co-authors that they can add their ORCIDs to their accounts and that they must do so prior to acceptance.

Version 1:

Decision Letter:

Dear Professor Jenkins,

Your revised manuscript titled "Revisiting Ice shelf Basal Melt Parameterisations for Ice-Ocean Boundary Layers over dynamically stable Pycnoclines" has now been seen by our reviewers, whose comments appear below. In light of their advice we are delighted to say that we are happy, in principle, to publish a suitably revised version in Communications Earth & Environment.

We therefore invite you to revise your paper one last time to address the remaining concerns of our reviewers. At the same time we ask that you edit your manuscript to comply with our format requirements and to maximise the accessibility and therefore the impact of your work.

EDITORIAL REQUESTS:

****Please take care to match our formatting and policy requirements. We will check revised manuscript and return manuscripts that do not comply. Such requests will lead to delays. ****

SUBMISSION INFORMATION:

OPEN ACCESS:

Communications Earth & Environment is a fully open access journal. Articles are made freely accessible on publication. For further information about article processing charges, open access funding, and advice and support from Nature Portfolio, please visit <https://www.nature.com/commsenv/open-access>

Link Redacted

Best regards,

Jennifer Veitch, PhD
Editorial Board Member
Communications Earth & Environment
orcid.org/0000-0003-2544-1243

Alireza Bahadori, PhD
Associate Editor
Communications Earth & Environment
Consulting Editor
Communications Sustainability

REVIEWERS' COMMENTS:

Reviewer #1 (Remarks to the Author):

I have carefully reviewed the revised manuscript titled **"Revisiting Ice Shelf Basal Melt Parameterisations for Ice–Ocean Boundary Layers over Dynamically Stable Pycnoclines"** and compared it with the original submission. I am pleased to confirm that the authors have addressed all major concerns I raised during the initial review process, and the manuscript now meets the standards for publication in **Communications Earth & Environment**.

The revised version demonstrates substantial improvements in clarity, structure, and scientific rigor, notably:

- Clearer exposition of the physical processes: The authors have improved the explanation of the interplay between friction velocity, turbulent transfer, and the suppression of vertical mixing under stable pycnoclines. These clarifications make the work significantly more accessible and impactful to a broader geophysical modeling audience.

Strengthened observational context: The revised manuscript includes additional references, further supporting the relevance of the proposed parameterisation in real-world settings such as the Thwaites Glacier grounding zone.

Enhanced parameterisation framework: The authors have expanded and refined the analytical formulation, offering a more transparent and reproducible methodology. The validation against a broad model ensemble is convincing and the derived relationships (e.g., equations 8–12) are well-supported by data.

Improved figures and presentation: All figures have been revised for clarity, with consistent notation and improved legends. The improvements aid significantly in the interpretation of key results.

Given these improvements, I find that the revised manuscript makes a valuable contribution to the field of ice-ocean interactions and ocean model development. It presents a physically sound and computationally efficient parameterisation approach that is highly relevant for large-scale ocean and climate modeling.

I therefore recommend acceptance of the manuscript for publication without further revision.

Reviewer #2 (Remarks to the Author):

My overall impression of this paper is still positive. Figures and conclusion are clearer with this first round of revision although some parts of the manuscript remain a little too technical to read for non-specialists.

I still have a few moderated comments, detailed below:

L22-23: sentence unclear. By reading the paper, I understood that the quick fix to minimise melt rate overprediction, at least for ocean modeller, is to reduce the size of the prescribed boundary layer depth (named h in the manuscript), which is linked to the grid resolution.

L235-236: sentence unclear. It is not easy to understand the generic term 'data'. Does this term refer to calculated or parameterised output?

L237-238: This sentence is redundant with the previous one.

L310: It is difficult to understand the structure of this section.

From L311 to L371, authors compared their parameterisation with the one from Yung et al. (2024) and observations. Then, from L372, authors discussed physics that can be introduced to make their parameterisation more complex. Perhaps, this new paragraph should be introduced in another section. For example, "toward a more complex framework". If this option is not chosen, a transitional paragraph is necessary between L371-372.

L378: Authors should add a comma after "lateral fluxes".

L417: "suppress" instead of "suppresses".

L573: unexpected line.

Reviewer #3 (Remarks to the Author):

This is my second review of the paper by Jayasankar and Jenkins which describes a simple parameterisation for calculating the rate of basal melting in the ice-ocean boundary layer beneath nearly horizontal ice shelves where the boundary layer is bounded by a sharp pycnocline. The authors have addressed all my concerns, and I now consider the paper ready for publication. I have noted some very minor corrections in the attached PDF.

** Visit Nature Portfolio's author and referees' website at <http://www.nature.com/authors> for information about policies, services and author benefits**

Revisiting Ice shelf Basal Melt Parameterisations for Ice-Ocean Boundary Layers over dynamically stable Pycnoclines

T. Jayasankar^{1,2}, A. Jenkins¹ *

¹University of Northumbria, Newcastle upon Tyne, United Kingdom.

²Now at Norwegian Polar Institute Fram Centre, Tromsø, Norway.

Corresponding author: Adrian. Jenkins (adrian2.jenkins@northumbria.ac.uk)

Geography and Environmental Sciences

University of Northumbria, Newcastle upon Tyne, United Kingdom

Reply to review comments by Authors

The authors thank the reviewers for their thoughtful and valuable comments towards the improvement of this manuscript. We apologise for the inconvenience caused by flaws in the manuscript related to figures, their colour schemes, legends etc that could have been avoided. We also thank the reviewers for rightly pointing them out.

Reviewer #1

The manuscript discusses the revision of basal melt rate parameterizations under ice shelves, with a particular focus on cases where stable density stratification (pycnocline) suppresses vertical heat mixing. Using an Ice-Ocean Boundary Current Model (IOBCM), the authors analyze the dynamics of the boundary layer (BL) over a stable pycnocline and compare calculated melt rates with those parameterized in ocean models. The manuscript highlights that conventional parameterizations tend to overestimate melt rates under conditions of weak currents and strong stratification, as observed in the grounding zone of Thwaites Glacier (Davis et al, 2023). The authors propose a new physics-based parameterization framework that considers the crucial role of limited diffusivity in the pycnocline and the balancing of heat fluxes. This approach aims to improve the representation of basal melting in low-resolution ocean models, contributing to a better understanding of ice shelf stability and sea level rise projections.

In general, I found the reading of the manuscript difficult. Considering the importance of the topic, I would have preferred a presentation of the results that was or more descriptive or with a better description of the many approximations and assumptions used to derive the equations. In a short paper (such this one), the contents and the passages, that sometimes are not intuitive, are hard to follow. The manuscript can certainly be considered for publication, but for this type of manuscript (compact), my suggestion is to move as much as possible the description of the derivation of the equations into the methods/supplementary material and

then give more space to the description of the consequences and content of the figures. Indeed, the paper appears somehow very technical.

Therefore, it may not be easily accessible to a reader who are not fully versed in the theoretical aspects of these processes. Thus, my main suggestion is to try to make the paper easier to read.

We have revised the manuscript based on the review comments and we hope this has improved the clarity. While the reviewer has suggested moving all the equations to the methods sections, we believe that a simple and concise presentation of the physics-based parameterization framework that is the focus of this manuscript requires the use of at least some equations, particularly to clarify similarities to and differences from other related work. In the revised manuscript we have devoted more space to the explanation of processes (including in the expanded methods section) that are represented by the basic equations presented in the main text. We hope that the reviewer finds this approach acceptable and the presentation now sufficiently clear.

Minor points

Lines 109-111: are the values used for U_{FF} realistic for the ocean? Please explain why did you use this values. Furthermore, it could be useful to have an indication about how far from the IOI this velocity can be observed.

We thank the reviewer for the eye for detail. The magnitudes were wrong. We have rightly changed them to, $U_{FF} \cong 0.35$ m/s and $U_{FF} \cong 0.14$ m/s. The values were chosen such that there are noticeable changes to viscosities and diffusivities while the behaviour for faster and slower currents can be differentiated.

Line 148: is 30m a realistic estimate of the BL depth? It looks like that this value can be observed only in very high turbulent regimes.

Prescribing a BL depth itself is far from reality, and the choice of depth is governed by the coarser grid resolution commonly used in ocean models. In the standard experiments of Mathiot et al. (2017), used to study melt rate parameterization in the NEMO ocean model, the BL depth prescribed throughout is 30 m.

Line 156: how did you chose the depth (30 m) of the BL?

The purpose of Fig. 2 is to demonstrate the potential for overprediction of melt rates by the current parameterization when far-field currents are slower. To effectively demonstrate this with a minimal number of data points, it's convenient to use a boundary layer (BL) depth large enough to yield a sufficiently wide friction velocity window. Conveniently, a 30m BL depth, which is also prescribed by Mathiot et al. (2017) in their NEMO ocean model experiments, perfectly fits this criterion.

Line 156: is it correct the figure indicated at the end of the line (Fig 1e)?

We thank the reviewer for the eye for detail. It is Fig 1d and has been changed in the manuscript.

Line 159: is it correct to use the term spike? In my understanding, a spike is an anomalous value. Here it looks like that it is just and higher value.

The term 'spike' is used twice in the manuscript and in the first place it is replaced by the term 'considerable overprediction' and in the second place it is replaced by the word 'overprediction'.

Lines 185-186: considering that the typical depth of BL is less than 30 m according to the literature (see comment above for line 148) wasn't this consideration already somewhat expected?

Yes, it is expected, but the choice is dependent on the ocean model grid and probably other factors. While Mathoit et al., (2017) used a default value of 30m, that choice isn't explicitly justified in their manuscript.

Lines 228-229: In equation 10 the authors introduced K_0 but this is the first time they used it. In the method section it is indicated that it is the effective thermal diffusivity. It would be useful to indicate what it represents here too or to refer to the Methods equation 22.

We apologise for this missing this. It is now introduced in the main text as follows (L244):

In the above equation, apart from the effective thermal diffusivity at the IOI (eq. 22), K_0

Line 248: Is approximate the right verb to use here? In which sense $TT*\delta\delta$ is approximated by $u*0$?

We have replaced 'approximate' with the 'derived'.

Line 263: You just introduced $u*_{FF}$ assuming a linear profile for currents within the BL. Can you explain which could be the error in $u*_{FF}$?

The argument based on a linear profile was a wrong interpretation from our side, and we thank the reviewer for drawing our attention back to this. We have rephrased as follows (L265-L271):

A simple approximation for u_{*0} , u_{*FF} , can be expressed as follows:

$$u_{*FF} = \frac{1}{2} |U_{FF}| \sqrt{C_d}, \quad (13)$$

Note that the above parameterisation for u_{*FF} is case specific and based on a quasilinear relationship (Fig 1e and 4a). While the constant of proportionality of 1/2 is best suited for the HTC scheme (at least for the range of $\sqrt{C_d}$ values used, between 0.05 and 0.12) it reduces further for the PP mixing scheme (Fig 1e). The parameterisation of friction velocity is thus mixing scheme dependent.

Lines 258 to 274: I suggest the authors to improve the clarity of this part of the manuscript.

We have updated Fig 4 and rephrased the section as follows (L277-L296):

Predicted values of $T_{*\delta}$ are compared with the actual values (y-axis) in Fig 4b. When u_{*0} is used as a predictor, the values agree well and fall mostly along the zero-error line (black dots). When, u_{*FF} is used as a predictor, for lower values of u_{*FF} , where $T_{*\delta} \approx T_{*a}$, the predicted values fall on the zero-error line (coloured dots in Fig 4b, most noticeable for the case where $T_{*a} = 4$ with $\sqrt{C_d} = 0.005$ and $\sqrt{C_d} = 0.12$). With an increase in u_{*FF} , $T_{*\delta}$ deviates further away from the zero-error line before eventually falling back on the line when the limiting lower value is approached. Interestingly, when u_{*FF} is overpredicted $T_{*\delta}$ is underpredicted. Conversely, when u_{*FF} is underpredicted (a small sample in Fig 4a for higher values of u_{*FF} for $\sqrt{C_d} = 0.12$) $T_{*\delta}$ is over predicted (a small sample in Fig 4b corresponding to $\sqrt{C_d} = 0.12$). Considering that melt rates are the product of u_{*FF} and $T_{*\delta}$, the inverse relationship between these variables discussed above results in cancellation of error in one variable by the error in the other, at least partially. This could be the reason why melt rate predictions using eq. 15 are not severely affected by this error except for extreme values of Γ (Fig 4c). Nonetheless, once $T_{*\delta}$ is close to its limiting value (which happens quickly), it remains more or less the same despite changes in u_{*FF} . Hence, if m_{Kb} is obtained by fitting predicted melt rates to data, errors will be confined to a regime of weaker currents, or lower u_{*0} . Since melting is also low for low u_{*0} , satisfactory prediction can be expected. Since, m_{Kb} governs warming by mixing across the pycnocline, it only affects the limiting value of $T_{*\delta}$, as illustrated by the sensitivity of $T_{*\delta}/T_{*a}$ to m_{Kb} for $\Gamma = 0.006$ (Fig 4d). Hence, just by tuning m_{Kb} , parametric models can be fitted to numerical models or observations for this special case of BL-DSP.

Line 265: From what I see looking at figure 4c, the highest errors occur when $\Gamma = 0.006$ that is a value in the middle of the range used.

We respectfully disagree with this. While results for $\Gamma = 0.006$ lie mostly within the 30% error line, the results for $\Gamma = 0.001$ and $\Gamma = 0.1$ lie further from zero error line and most are outside the 30% error line. $\Gamma = 0.001$ and $\Gamma = 0.1$ are introduced to test these extreme cases and to see how well they agree with, or deviate from, the physics captured by eq. 8. Also, better parameterization for melt rates could be achieved if data points pertaining to each Γ value were treated as separate cases. However, the focus of this manuscript is not improving the accuracy of the a specific implementation of the parameterized IOBCM but introducing the new physics-based parametrization framework. The ultimate aim remains better parametrization of observations and models in general, informed by the IOBCM, rather than parameterization of the IOBCM itself.

Line 299: What LES does mean?

The expansion “Large Eddy Simulation” is added to the manuscript.

Line 320: We have very little information and observations in the cavity, so the authors may be right when they say that temperature observations taken in a borehole may be close to those in the BL. I wonder what the distance from the BL at which T^* is measured and whether using different values the estimated m has more realistic values.

We agree with the sentiment behind the reviewer’s comment. The success of any parameterization in reproducing observed melt rates depends on whether the observations are made at an appropriate depth below the IOI to be consistent with the physics embedded within the parameterization, a point that often seems to be overlooked. In our manuscript we

explicitly show the different results obtained when taking temperatures from above or below a dynamically stable pycnocline and show that our new parameterization framework can account for that difference. Based on the results obtained with that new framework, we suggest that the observations in question were from below the pycnocline, but we have no more information to support that suggestion beyond what is available in the published literature.

Lines 340 to 342: A clearer explanation might help the reader to better understand this concept.

We have rephrased it to the following (L377-L386):

Hence, while lateral advection drives melting, the pathway to the IOI is through the pycnocline. Observations by Davis et al., (2025) from depths below the pycnocline validate this. Conveniently, in the IOBCM, T_{*a} which in the real world is maintained by lateral fluxes is maintained by fixing its value. Hence, there is an implicit assumption that lateral advection below the pycnocline is the ultimate source of heat for melting, but that melting is regulated by diffusion through the pycnocline. That assumption necessitates the redirection of primary focus from the BL and the IOI to mixing across the pycnocline through m_{kb} in eq. 15. Similarly, a recent numerical study of ice-ocean BL mixing reported that while fluxes across the IOI could be captured with relatively coarse near-interface vertical grid resolution, finer resolutions in the regions of entrainment were critical (Burchard et al., 2022).

Comments on figures.

Figure 1 a,b,c i would draw the line representing the boundary layer depth with a different colour to avoid confusion with the curve representing the profiles of u , k and v in for the high far field velocity case.

We have updated as suggested. Thank you.

Figure 2 : there are no indications in the figure caption about what the different colours of the dots represent. The legends in b4 and d4 are not enough.

We have updated as suggested. Thank you

Figure 3: please check the format of the letters used to identify the subplots. For instance, c) is not in bold.

Thank you for the eye for detail. It is updated.

Figure 4: please indicate for panel a what does the bold, continuous line represent. I guess the zero error line, but you should indicate it.

Zero error line is now mentioned in the Figure caption.

In the Dataset section Table 1, please explain what the parameter “Size” represents.

Size here means the total number of simulations performed using the IOBCM for generating respective data. It has now been rephrased as “Total no. of simulations”. Thank you.

Review

Summary – Key results

This study presents a new formulation of a basal melt parameterisation for ice-ocean boundary layers to overcome current inconsistencies between parameterisation implemented in oceanic models and observations in relatively flat areas.

The revised parameterisation is based on the results of an Ice-Ocean Boundary Current Model (IOBCM) that parameterises turbulent processes using a mixing length approach. The authors first check that the model captures the desired physics.

Then, they compare the results of the model with the current parameterisation implemented in ocean models. Results suggest that when h (the depth of model first grid point) is greater than the depth of the boundary layer, the current parameterisation overestimates basal melting. Reducing the value of h could be a first step towards a more realistic estimate of melting in low-slope areas. By assuming that warming by mixing across the pycnocline is the most important element, authors propose a new parameterisation framework and compare its performance with the IOBCM model. This parameterisation ensures that the observed physics in flat areas is properly represented, and seems relatively easy to use. The results are promising, although the parameterisation remains under development and needs to be further validated on more complex geometries.

Comment & suggestions

My overall impression of this paper is fairly positive. The paper sheds new light on ice-ocean interactions under ice shelves and proposes several ways of correcting a basal melt parameterisation in order to overcome inconsistencies between parameterisations and observations. The methods are clearly explained and the figures are relevant for the reader's understanding. This manuscript will likely be a significant contribution with revision and clarification about (i) recommendations for the modelling community and (ii) a wider discussion on the impact of the corrected process vs. the other processes neglected in the basal melt parameterisation. I made a few comments. I split the comments into two categories: major (request) and minor (suggestion). The aim of this request is to ensure that the reader clearly understands what corrections can be implemented now in the ocean model and what sources of uncertainty remain and still need to be investigated.

Major points

- I think the abstract/conclusion might be clearer:
 - abstract: I think the authors should add in the abstract what the quick fix consists of, as it is

one of the conclusions of the study. That's mean, reducing the distance between the first grid

We have modified the abstract as follows and we hope this is sufficient. Thank you (L23):

We suggest a quick fix for this (prescribing low melt rates for slower currents).

point and the base of the ice shelf. - conclusion: recommendations on the new formulation of the basal melt parametrisation are missing. Parameter values or indications on how to choose them should be provided and the remaining limitations specified. The authors focus on the calibration of the value of m_k , but it is not clear to me what values/range of values to consider for γ , K_b , C_d and d_{pyc} . • I'm not an expert in basal melt parameterisation but I feel that a discussion about the impact of the results presented here is missing, and that the authors should attempt to put them into a more general context:

The following section has been added as the last paragraph of the Conclusion (L459-L470).

The versatility of the parameterisation framework lies in the two-way physics-based control on ice-ocean heat exchange in eq. 15 through fluxes across the IOI and the pycnocline. Control of heat flux across the IOI is achieved by means of the turbulent transfer coefficient, Γ . The same across the pycnocline is achieved through parameters m_{Kb} (mixing across the pycnocline) and d_{pyc} (pycnocline depth). In this introductory work on the parameterisation framework, the simple case of a flat IOI allows usage of constant values for these parameters. For more complicated settings of BL-DSP, such as when the IOI is sloped, both should be represented as functions of slope. While that requires further analysis of observations and model output, the framework presented here lays a useful foundation. Owing to the simplicity of eq. 15, modelers can try fitting available data to it and make informed deductions towards improvement of these parameters including Γ . The underlying framework provided is versatile, allowing for subsequent validation and expansion, which will be explored in follow up work.

- since the authors stress the importance of basal melting/geometry feedback, what is the validity of their new parameterisation in a context of changing geometry? (for example, are the value of calibrated parameters still valid in another geometry? for sloped IOI?)

In this manuscript our focus is the general parameterization framework and its application in the simple case of a flat IOI. For more general geometries, the parameters presented as constants for this special case of flat a IOI will become functions of slope. Thus, the melt-geometry feedback will be captured. This is beyond the scope of the present manuscript.

The authors focus on the calibration of the value of m_k , but it is not clear to me what values/range of values to consider for γ , K_b , C_d and d_{pyc} .

The value of K_b is a default value of 10^{-5} m²/sec and is now included in the manuscript. The range of C_d values considered is also given in the manuscript along with the observed d_{pyc} , although because the parameterization is based on u^ , which is a function of C_d , we do not focus on C_d . While, the values given in the manuscript are applicable for emulating the*

IBOCM as configured in the study, the same approach could be applied to observations or another model. This aspect is now stressed in the manuscript, as follows (L295-L300):

Hence, just by tuning m_{kb} , parametric models can be fitted to numerical models or observations for this special case of BL-DSP. We limit the scope of this manuscript to that special case. However, for more general geometries, instabilities associated with sloped IOIs enhance vertical mixing, which causes an increase in both m_{kb} and d_{pyc} with slope. Hence, both these parameters would have to be represented as a function of slope to account for slope dependant instabilities/mixing.

- I understand that the generalisation to more complex and realistic geometry is planned for a future paper. Nevertheless, I think that a general discussion is missing on the absence of some processes that could also influence (in particular decrease) the melting under Thwaites.

For a more realistic geometry that is sloped, there will be a significant influence of Γ . Also, both m_{kb} and d_{pyc} will increase due to instabilities associated with vertical mixing. So, both the parameters have to be expressed as a function of slope as well. This is now clearly mentioned in the manuscript and is presented in our response to the queries above. However, derivation of functions for these variables itself is a subject for another manuscript. Since, this paper is about the versatility of the parameterisation framework demonstrated through a flat IOI, we present these parameters as constants.

We thank the reviewer for suggesting the following references. These references certainly have improved the understanding of the authors as well as opened new research questions and possibilities.

For example:

- the impact of the tide in vertical mixing (Jourdain et al. 2019)

The following section is added to methods (L542-L573):

Decomposition of melt rate change

The underlying physics of the parameterised equation allows us to draw informed deductions. For example, Jourdain et al., (2019) ran ocean model experiments for Amundsen Sea ice shelves with and without tidal forcing. They analysed the influence of tides on melt rates by decomposing the change in melt rate into dynamical and thermodynamical components. A similar decomposition of eq. 6 yields the following equation for the change in melt rate:

$$\Delta\dot{m} = \left(\frac{c}{L_i}\right)\Gamma\Delta u_{*0}T_{*\delta} + \left(\frac{c}{L_i}\right)\Gamma u_{*0}\Delta T_{*\delta} + \left(\frac{c}{L_i}\right)\Gamma\Delta u_{*0}\Delta T_{*\delta}, \quad (27)$$

In the above equation, the first term denotes the dynamical term, where Δu_{*0} represents the change in friction velocity due to the inclusion of tides. Similarly, the second term

denotes the thermodynamical term where $\Delta T_{*\delta}$ represents the change in BL thermal driving due to the inclusion of tides. The last term is referred to as a covariational component. The above decomposition ignores the strong coupling between u_{*0} and $T_{*\delta}$ due to the growth of the BL. Hence, Jourdain et al., (2019) attribute the relatively large contribution of the thermodynamic and covariational components (nearly one third of the dynamical) to non-local processes or tide-induced cooling of the continental shelf. A similar decomposition considering the coupling between variables can be done by means of eqs 6, 8 and 9. From eq 6:

$$\frac{d\dot{m}}{du_{*0}} = \left(\frac{c}{L_i}\right) \Gamma T_{*\delta} + \left(\frac{c}{L_i}\right) u_{*0} \Gamma \frac{dT_{*\delta}}{du_{*0}}, \quad (28)$$

Multiplying the above equation with Δu_{*0} :

$$\Delta \dot{m} = \left(\frac{c}{L_i}\right) \Gamma T_{*\delta} \Delta u_{*0} + \left(\frac{c}{L_i}\right) u_{*0} \Gamma \frac{dT_{*\delta}}{du_{*0}} \Delta u_{*0}. \quad (29)$$

Since $\frac{dT_{*\delta}}{du_{*0}} \Delta u_{*0} = \Delta T_{*\delta}$, the equation is similar to eq. 6 wherein first term can be inferred as the dynamical component and the second as the thermodynamical component. From eq. 8 and 9 (also resubstituting $T_{*\delta}$ back after differentiation):

$$\frac{dT_{*\delta}}{du_{*0}} = -T_{*\delta} \frac{d_{pyc} K_b \Gamma}{(d_{pyc} u_{*0} \Gamma + K_b + m_{Kb} u_{*0})(K_b + m_{Kb} u_{*0})}, \quad (30)$$

From eq. 29 and 30, a ratio of thermodynamical to dynamical terms can be expressed as follows:

$$R_{T/D} = - \frac{u_{*0} d_{pyc} K_b \Gamma}{(d_{pyc} u_{*0} \Gamma + K_b + m_{Kb} u_{*0})(K_b + m_{Kb} u_{*0})}. \quad (31)$$

In the main text the following is added; Fig 5 is also updated with an additional plot (L392-L410):

While the inclusion of tides in ocean models increases the melt rate through a dynamical component related to increased friction velocity, a third of that increase is negated by a thermodynamical component (see methods, eq. 27-30) associated with cooling of the BL (Jourdain et al., 2019). Although the tendency of the BL to cool with increasing friction velocity is implicitly accounted for, the effect has often been overlooked, leading to the

conclusion that non-local processes cause the BL cooling (Jourdain et al., 2019). Here we investigate the ratio of the thermodynamical to dynamical components ($R_{T/D}$) of the melt rate change with changes in friction velocity within our parameterisation framework (Fig 5b; the relevant parameters are adopted from Jourdain et al., 2019). It is possible for the thermodynamical component associated with the local mixing processes considered in our parameterised IOBCM to negate over half of the dynamical component, so they could certainly account for the third reported by Jourdain et al. (2019). However, the impact of the thermodynamical component reduces for higher values of m_{Kb} , and hence with higher values of IOI slope, since increased slope enhances mixing across the pycnocline.

Observations under Thwaites glacier show that melting increases with the slope of the IOI (Schmidt et al., 2023). In principle, such geometrical control of melting could be accounted for by expressing m_{Kb} as a function of slope based on observations and/or models more complex than IOBCM. However, such a development is beyond the scope of the present study, where we seek to present our new parameterisation framework and demonstrate its versatility

- the impact of accounting for heat conduction into the ice (Wiskandt et al. 2024)

The following section is added to methods (L528-L541):

Heat conduction into the ice

Additionally, along with turbulent heat transport across the IOI, it is possible to include heat conduction into the ice shelf in eq. 7. Following a simple model (eq. 1 in Wiskandt and Jourdain, 2024) for heat conduction into the ice shelf, eq. 7 can be modified as follows:

$$D_{BL} \frac{dT_{*\delta}}{dt} = u_{*0} \Gamma (0 - T_{*\delta}) - \frac{\rho_i c_i K_i}{\rho c} \left(\frac{\partial T_i}{\partial z} \right) + \frac{K_{pyc} (T_{*a} - T_{*\delta})}{d_{pyc}}, \quad (25)$$

In the above equation ρ_i , c_i , and K_i refer to density of ice, specific heat capacity of ice and thermal conductivity of ice respectively, ρ denotes density of sea water and T_i denotes far-field temperature inside the ice shelf. Assuming a steady $T_{*\delta}$, this leads to the following expression for $T_{*\delta}$:

$$T_{*\delta} = \frac{K_{pyc} T_{*a} - d_{pyc} \frac{\rho_i c_i K_i}{\rho c} \left(\frac{\partial T_i}{\partial z} \right)}{d_{pyc} u_{*0} \Gamma + K_{pyc}}. \quad (26)$$

A model that also captures heat conduction into the ice shelf can be trained using the above equation. Following the parameterisation framework in this manuscript, K_{pyc} and d_{pyc} can be parameterised by fitting the model results to the above equation.

In the main text the following is added (L390-L392):

The versatility of the parameterisation framework also allows inclusion of other contributions to the heat flux across the IOI. For example, heat conduction into the ice shelf (Wiskandt and Jourdain, 2024) can be simply incorporated (Methods eq. 25 and 26).

- the impact of roughness like rift, crevasses (Poinelli et al. 2023,

The following section is added to methods (L386-389):

While our parameterisation is based on a constant T_{*a} , more complicated settings, such as when dense waters from inside rifts sink and cool down the ambient waters (Poinelli et al., 2023), can also be accounted for using eq. 15 by introducing functions adjusting T_{*a} .

Schmidt et al. 2023)

While Schmidt et al., 2023 is cited in the manuscript for the general conclusion that melt rate is dependent on slope, the observations were made in a region characterised by heterogeneities in ice shelf basal topography, such as crevasses, having horizontal scales comparable with the BL depth. Our IOBCM is not appropriate for estimating melt rates in such a region, so we would not anticipate being able to reproduce the melt rate variability without the use of a more complex model.

- the impact of advection vs. vertical turbulent heat fluxes (Davis et al. 2025)

In the main text the following is added (L373-L383):

The stratification under a low IOI slope considered here implies quasi-horizontal isopycnals and slow flow, implying negligible lateral heat advection within the BL. Below the pycnocline, vertical mixing is minimal and the heat loss to BL through the pycnocline should therefore be supplied through lateral heat advection below the pycnocline, controlled by the large-scale sub-ice circulation. Hence, while lateral advection drives melting, the pathway to the IOI is through the pycnocline. Observations by Davis et al., (2025) from depths below the pycnocline validate this. Conveniently, in the IOBCM, T_{*a} , which in the real world is maintained by lateral fluxes, is maintained by fixing its value. Hence, there is an implicit assumption that lateral advection below the pycnocline is the ultimate source of heat for melting, but that melting is regulated by diffusion through the pycnocline. That assumption necessitates the redirection of primary focus from the BL and the IOI to mixing across the pycnocline through m_{kb} in eq. 15.

Minor points

L13: “basal melt rates” instead of “basal melts rates”

Corrected.

L14 and 54: “suppress” instead of “suppresses”

Corrected.

L21-22: sentence unclear. What are the quick fixes? I think the authors should add here what this quick fix consists of, as it is one of the conclusions of the study.

We have modified the abstract as follows and we hope this is sufficient. Thank you (L23):

While we suggest a quick fix for this (prescribing low melt rates for slower currents).

L52: The authors can also cite Schmidt et al. (2023) as the conclusion is similar.

Reference cited.

L56-57: It would be interesting to calculate the slope under the ice shelves, using a dataset such as BedMachine for example, just to assess whether the low slopes under the ice shelves are common and whether many ice shelves could be affected by this overestimated melting.

We agree that this must be investigated and should be investigated in detail using ocean models like NEMO. As you would understand we would like to keep such detailed works beyond the scope of this work considering the work stemming mostly out of a theoretical standpoint for a special case of flat Ice Ocean interface. Considering the availability of limited manhours, we have at our disposal such detailed investigation is challenging for us presently. Hence, we request the reviewer's kind consideration on this matter

Figure 1: Would it be possible to make the labels a little bigger or bold to make them easier to read?

We have updated as suggested.

I think the unit for the y-axis is missing for Fig 1g (DBL in m?)

All units are added.

The authors could also be a little more precise about the legend in Figure 1a. HTC or PP is indicated for the red and blue curves. The authors could add an additional qualifier to more easily differentiate the difference between the two curve colours (like Uff value?).

We have updated as suggested.

The choice of highlighting the depth of the boundary layer in the same colour as the profiles is a little confusing in Fig 1abc.

We have updated as suggested.

In Fig 1e, it is not clear that the authors are using two values for the Cd coefficient. Would it be possible to clarify this on the figure?

We have updated as suggested.

L150: Equation 25 is referred to several times in the body of the text. Wouldn't it be a good idea to introduce it in the body of the text?

We have updated as suggested. Eq 25 in the first draft is now included in the main text as eq 6.

Figure 2: It would be easier to understand if you put columns 1 and 3 next to each other and columns 2 and 4 next to each other, with each of the two columns having the same vertical amplitude for melt rate. It would also be useful to remind the reader what m and m_h correspond to in the legend of the figure:

We have updated as suggested.

“Comparison of calculated (m) and parameterised (m_h , BLM approach) melt rates...”

We have updated as suggested.

L222: sentence unclear. Which data are you talking about?

The sentence is rephrased as follows (L236):

Eq. 8 (black lines in Fig 3a) fits the data (coloured dots in Fig 3a) well when m_{Kb} is set to 0.0007.

Figure 3: Why is the colour code changing between figures 3a and 3c? A single colour per gamma would be more appropriate.

Color code is updated to be uniform.

L289-290: sentence unclear.

The sentence is rephrased as follows and we hope it is clear now (L316-L317):

Currents ranging from 2.0 to 5.4 cm/s were observed under the ice shelf and this yields melt rates between 14.0 and 30.0 m/y using the canonical model for basal melting (see Davis et al., 2023) versus observed rates of 2.0 to 5.0 m/y.

L292-294: the authors could specify that they are referring to figure 4ef for dots and lines.

We have updated as suggested. Thank you.

Figure 5: As Georges VI and Amery are superimposed, it is not clear whether the parameterisation does not fit Amery or Georges V melt rate. What could explain the differences between the parameterization and the boreholes values for Georges VI or Amery?

The figure is updated such that values for Georges VI and Amery can be distinguished now.

The melt rate is overpredicted for George VI. The more conservative melt rate (blue dot) for George VI is a melt rate we got when we used 0.025 for m_{Kb} . However, 0.025 is just a hypothetical value that we have applied uniformly for all locations. The high melt rate is

predicted for George VI when m_{Kb} set to 10 (flat line). A high value of m_{Kb} means there is negligible cooling of the BL and there is no stratification such that the ambient thermal driving is available at the IOI. This should result in an unrealistically high melt rate (as suggested by the flat line). We also expected a similar response for all the other locations. However, they were all matching well. That is why we suspect that the ambient temperature used in the calculation of melting at other locations may be from locations within their BL and not from a depth below the pycnocline. From Kimura et al. (2015; <https://doi.org/10.1175/JPO-D-14-0106.1>) it can be confirmed that the values taken for George IV are from a depth below pycnocline. Hence when such a high m_{Kb} is used in eq. 15, unrealistic overprediction as given by the flat line is expected.

L343: The following paragraph is more a conclusion than a discussion. The previous part that focuses on the comparison of the parameterisation with observation and Yang's parameterisation is more similar to a discussion. Should the name of the paragraph be changed?

We agree. We have updated as suggested.

L464: extra parenthesis

We have updated as suggested. Thank you

Miscellaneous

Appropriate use of statistics and treatment of uncertainties: yes.

Conclusion: the conclusions and data interpretation are robust, valid and reliable.

References: yes

Clarity and context: the abstract is fairly clear and accessible (see comment on L21-22).

Introduction

and conclusions are appropriate.

Inflammatory material: no

References for this review (not cited in the paper) All references are appropriately cited

Davis, P. E., Nicholls, K. W., Holland, D. M., Schmidt, B. E., Washam, P., Castro, B. F., ... & Makinson, K. (2025). Lateral Fluxes Drive Basal Melting Beneath Thwaites Eastern Ice Shelf, West Antarctica. *Geophysical Research Letters*, 52(3), e2024GL111873.

Jourdain, N. C., Molines, J. M., Le Sommer, J., Mathiot, P., Chanut, J., de Lavergne, C., & Madec, G. (2019). Simulating or prescribing the influence of tides on the Amundsen Sea ice shelves. *Ocean Modelling*, 133, 44-55.

Poinelli, M., Schodlok, M., Larour, E., Vizcaino, M., & Riva, R. (2023). Can rifts alter ocean dynamics beneath ice shelves?. *The Cryosphere*, 17(6), 2261-2283.

Schmidt, B. E., Washam, P., Davis, P. E., Nicholls, K. W., Holland, D. M., Lawrence, J. D., ... & Makinson, K. (2023). Heterogeneous melting near the Thwaites Glacier grounding line. *Nature*, 614(7948), 471-478.

Wiskandt, J., & Jourdain, N. (2024). Brief Communication: Representation of heat conduction into the ice in marine ice shelf melt modeling. *EGUsphere*, 2024, 1-10.

Reviewer #3

The paper by Jayasankar and Jenkins describes a simple parameterisation for calculating the rate of basal melting in the ice-ocean boundary layer beneath nearly horizontal ice shelves where the boundary layer is bounded by a sharp pycnocline. Their model builds on previous modelling of the ice-ocean boundary layer and represents a significant advancement in the field of ice-ocean interactions. Their model can represent the competing effects on basal melting of strong density stratification at the ice base and mixing driven by far field flow speeds. Unlike conventional models of basal melting which overpredict the basal melt rate in the presence of strong stratification and slow far-field current speeds, the model of Jayasankar and Jenkins correctly predicts a highly suppressed rate of melting. Their model can recreate observed rates of basal melting beneath the eastern Thwaites glacier grounding zone, demonstrating that their model has wide applicability. The work presented in this paper will have significant implications for modelling ice-ocean interactions and therefore I recommend it as suitable for publication. There were a few locations in the text and figures, however, where I felt that a little extra clarity or improved presentation was required. Once these areas have been addressed the paper will be ready for publication. I have highlighted my areas of concern, along with minor corrections that are required, in the attached pdf. **

We have incorporated all the corrections/changes suggested by the reviewer. Please note that the corrections made are not explicitly listed in this reply document. However, the concerns raised are addressed below.

L117-118: Please add all units to axes labels

We have updated as suggested. Thank you

L121-122: Please clarify with a legend in panel e which markers refer to which drag coefficient

We have updated as suggested. Thank you

L 125: Please give a value of the Ekman depth. Otherwise there is no way the actual depth in physical units of the BL can be determined from Figure 1.

We have updated as suggested. Thank you

L136-137: I think this statement needs clarifying for the reader as it is not immediately obvious. Is it the case that the higher heat flux causes more melting, which cause more cool fresh melt water to enter the BL and thus cause the thermal driving to decrease? If so, it would be worth making this explicit

We rephrased them as follows (L141-L147) :

Ice-ocean heat exchange becomes more efficient with increasing u_{*0} (Eq 24 in Methods) enhancing the melt rates but trapping the melted waters within the BL. While the growth of the BL with u_{*0} increases its thermal inertia, the increasing heat loss at the IOI with u_{*0} cools the BL. Hence, though counter intuitive, in both schemes, with an increase in u_{*0} , $T_{*\delta}$ decreases. Cooling of the BL with increasing currents has been reported in modelling studies although not explicitly attributed to fluxes across the IOI (Jourdain et al., 2019).

L156: How is this shown by Figure 1e which plots U_{FF} against u_{*0} ? Is there a typo here?

Yes, it is a typo. Thank you very much for the eye for details. We have corrected 'Fig 1e' to 'Fig 1d'.

L170: Please use the same axes limits for the different mixing schemes. i.e. a1) and c1), b1) and d1), etc. should have the same limits

We have updated the plot as suggested.

L273-274: In the regular three-equation melt rate parameterisation, the drag coefficient and turbulent heat and salt transfer coefficients can also be tuned and fitted to numerical models and observations to derive the correct melt rate. However it is this requirement for tuning that is often considered to be its greatest weakness as it can't be made to fit all regimes with a single set of coefficients. Can you comment (or make clearer) whether your model, which also appears to require tuning, is less susceptible to this tuning weakness,

Regarding the versatility of our parameterization framework we have added the following in the main text. We hope that this is sufficient (L359-L372):

Clearly, both the parameterisations of Y24 and eq. 15 have some merit. The former is built on a much more complete physical model, while the simplicity of the latter allows for flexible tuning. They are, however, quite different in concept. Y24, in common with other recent studies, focuses on improving the melt rate computed from eq. 6 through a consideration of the boundary layer physics subsumed into the turbulent transfer coefficient, Γ_{para} . In contrast, eq. 15 exploits the principle that for a horizontal IOI, lateral supply of heat must be negligible inside the BL, so $T_{*\delta}$ will adjust until vertical heat fluxes are balanced. Thus, unlike Γ_{para} , eq. 15 allows a physics-based control on the melt rate computation, exploiting the principal that, for a flat IOI, the heat flux through the pycnocline becomes the controlling factor, since diffusivity beyond the boundary layer will always be lower than within it. While eq. 15 remains simple, but offers a two-way control on the melt rate, one through the IOI by

means of Γ and the other through the pycnocline by means of m_{kb} and d_{pyc} . Conventional parameterisations lack this control since these fluxes are not explicitly distinguished in their derivations. Also, the influence of Γ inferred from Fig 3a could provide insight into the actual values of Γ once more observations are available.

and whether one value of m_{kb} can be used for widely varying ice-ocean boundary layer environments?

We expect it to be so for a given slope, because of the two-way control mentioned above block. We hope that others will be able to explore the wider applicability of the parameterization framework and specific parameter values through comparison with observation.

L335: How does this statement (“horizontal IOI, lateral supply of heat must be negligible”) relate to the recent paper of Davis et al. (2025): Lateral Fluxes Drive Basal Melting Beneath Thwaites Eastern Ice Shelf, West Antarctica.

Reviewer 2 asked a similar question. Hence, the following was added in the main text (L373-L383):

The stratification under a low IOI slope considered here implies quasi-horizontal isopycnals and slow flow, implying negligible lateral heat advection within the BL. Below the pycnocline, vertical mixing is minimal and the heat loss to BL through the pycnocline should therefore be supplied through lateral heat advection below the pycnocline, controlled by the large-scale sub-ice circulation. Hence, while lateral advection drives melting, the pathway to the IOI is through the pycnocline. Observations by Davis et al., (2025) from depths below the pycnocline validate this. Conveniently, in the IOBCM, T_{*a} , which in the real world is maintained by lateral fluxes, is maintained by fixing its value. Hence, there is an implicit assumption that lateral advection below the pycnocline is the ultimate source of heat for melting, but that melting is regulated by diffusion through the pycnocline. That assumption necessitates the redirection of primary focus from the BL and the IOI to mixing across the pycnocline through m_{kb} in eq. 15.

Revisiting Ice shelf Basal Melt Parameterisations for Ice-Ocean Boundary Layers over dynamically stable Pycnoclines

T. Jayasankar^{1,2}, A. Jenkins¹ *

¹University of Northumbria, Newcastle upon Tyne, United Kingdom.

²Now at Norwegian Polar Institute Fram Centre, Tromsø, Norway.

Corresponding author: Adrian. Jenkins (adrian2.jenkins@northumbria.ac.uk)

Geography and Environmental Sciences

University of Northumbria, Newcastle upon Tyne, United Kingdom

Reply to review comments by Authors

The authors thank the reviewers for their valuable comments towards the improvement of this manuscript. We are also grateful for the valuable learning experience gained while addressing the review comments.

Following the journal guidelines we have rephrased the manuscript title to the following.

Physics-based Parameterisation framework for Basal Melting in Ice-Ocean Boundary Layers over dynamically stable Pycnoclines.

Reviewer #1

The manuscript discusses the revision of basal melt rate parameterizations under ice shelves, with a particular focus on cases where stable density stratification (pycnocline) suppresses vertical heat mixing. Using an Ice-Ocean Boundary Current Model (IOBCM), the authors analyze the dynamics of the boundary layer (BL) over a stable pycnocline and compare calculated melt rates with those parameterized in ocean models. The manuscript highlights that conventional parameterizations tend to overestimate melt rates under conditions of weak currents and strong stratification, as observed in the grounding zone of Thwaites Glacier (Davis et al, 2023). The authors propose a new physics-based parameterization framework that considers the crucial role of limited diffusivity in the pycnocline and the balancing of heat fluxes. This approach aims to improve the representation of basal melting in low-resolution ocean models, contributing to a better understanding of ice shelf stability and sea level rise projections.

In general, I found the reading of the manuscript difficult. Considering the importance of the topic, I would have preferred a presentation of the results that was or more descriptive or with a better description of the many approximations and assumptions used to derive the equations. In a short paper (such this one), the contents and the passages, that sometimes are not intuitive, are hard to follow. The manuscript can certainly be considered for publication, but for this type of manuscript (compact), my suggestion is to move as much as possible the description of the derivation of the equations into the methods/supplementary material and then give more space to the description of the consequences and content of the figures. Indeed, the paper appears somehow very technical.

Therefore, it may not be easily accessible to a reader who are not fully versed in the theoretical aspects of these processes. Thus, my main suggestion is to try to make the paper easier to read.

I have carefully reviewed the revised manuscript titled *"Revisiting Ice Shelf Basal Melt Parameterisations for Ice–Ocean Boundary Layers over Dynamically Stable Pycnoclines"* and compared it with the original submission. I am pleased to confirm that the authors have addressed all major concerns I raised during the initial review process, and the manuscript now meets the standards for publication in *Communications Earth & Environment*.

The revised version demonstrates substantial improvements in clarity, structure, and scientific rigor, notably:

- Clearer exposition of the physical processes: The authors have improved the explanation of the interplay between friction velocity, turbulent transfer, and the suppression of vertical mixing under stable pycnoclines. These clarifications make the work significantly more accessible and impactful to a broader geophysical modeling audience.

Strengthened observational context: The revised manuscript includes additional references, further supporting the relevance of the proposed parameterisation in real-world settings such as the Thwaites Glacier grounding zone.

Enhanced parameterisation framework: The authors have expanded and refined the analytical formulation, offering a more transparent and reproducible methodology. The validation against a broad model ensemble is convincing and the derived relationships (e.g., equations 8–12) are well-supported by data.

Improved figures and presentation: All figures have been revised for clarity, with consistent notation and improved legends. The improvements aid significantly in the interpretation of key results.

Given these improvements, I find that the revised manuscript makes a valuable contribution to the field of ice-ocean interactions and ocean model development. It presents a physically sound and computationally efficient parameterisation approach that is highly relevant for large-scale ocean and climate modeling.

I therefore recommend acceptance of the manuscript for publication without further revision.

We thank the reviewer

My overall impression of this paper is still positive. Figures and conclusion are clearer with this first round of revision although some parts of the manuscript remain a little too technical to read for non-specialists.

I still have a few moderated comments, detailed below:

L22-23: sentence unclear. By reading the paper, I understood that the quick fix to minimise melt rate overprediction, at least for ocean modeller, is to reduce the size of the prescribed boundary layer depth (named h in the manuscript), which is linked to the grid resolution.

Reducing the size of the prescribed BL ‘ h ’ is recommended and it can definitely reduce the overpredictions according to our inferences. However, this reduction is still constrained by the size of the cells close to the IOI. The quick fix we suggest is to identify the conditions where overpredictions can be observed for slower currents. Then fix an upper cutoff value for melt rates when currents are slower. However, this now looks like a research direction than a simple quick fix. So, we have removed “quick fix” from the manuscript.

The abstract has also been rephrased to adhere to the journal's 150-word limit. The following text has been added to the updated abstract:

While reducing the prescribed BL depth can minimize this overprediction in ocean models, a better fix might be prescribing an upper melt rate limit for slower currents.

L235-236: sentence unclear. It is not easy to understand the generic term ‘data’. Does this term refer to calculated or parameterised output?

We have rephrased the sentence as follows:

The profiles for $\frac{T_{*a}}{T_{*a}}$ given by eq. 8 (black lines in Fig 3a) closely trace the true profiles given by IOBCM (coloured dots in Fig 3a) when m_{kb} is set to 0.0007. d_{pyc} is set to 8.0 m because for all cases, it is found to rise quickly with rising u_{*0} before plateauing at 8.0 m.

L237-238: This sentence is redundant with the previous one.

We thank the reviewer for eye for details. The sentence is removed.

L310: It is difficult to understand the structure of this section.

From L311 to L371, authors compared their parameterisation with the one from Yung et al.

(2024) and observations. Then, from L372, authors discussed physics that can be introduced to make their parameterisation more complex. Perhaps, this new paragraph should be introduced in another section. For example, “toward a more complex framework”. If this option is not chosen, a transitional paragraph is necessary between L371-372.

We agree with the reviewer and we have included a subheading as follows:

Potential for advanced framework and process studies.

L378: Authors should add a comma after “lateral fluxes”.

We have updated as suggested. Thank you.

L417: “suppress” instead of “suppresses”.

We have updated as suggested. Thank you.

L573: unexpected line.

We are not sure about this suggestion. There is blank line at 573. We assume that it will be removed at the editing stage before publishing.

Reviewer #3

This is my second review of the paper by Jayasankar and Jenkins which describes a simple parameterisation for calculating the rate of basal melting in the ice-ocean boundary layer beneath nearly horizontal ice shelves where the boundary layer is bounded by a sharp pycnocline. The authors have addressed all my concerns, and I now consider the paper ready for publication. I have noted some very minor corrections in the attached PDF.

We have incorporated all the corrections/changes suggested by the reviewer. Please note that the corrections made are not explicitly listed in this reply document. However, the concerns raised are addressed below.

L122: 1f is plotted as a function of gamma, but in the text you are discussing drag coefficient. I appreciate that the legend in 1f is required for panels g and h, but perhaps panel f can be plotted using drag coefficient to match the text, and the existing legend in f can be moved to panels g and h. I think this would make the figure match the text more clearly

We have updated as suggested. Thank you.

L242: More detail is needed in the main text about what equation 22 is and why it is being substituted into eq 8. Detail can be given in the methods, but it is currently impossible to follow from the main text why this substitution is being made.

We have rephrased the section as follows:

The diffusivity at the IOI expressed as $K_0 = u_{*0}\Gamma\delta$ (eq. 22), almost uniquely controls the BL thermal driving (Fig 3b) owing to its linear relationship with u_{*0} . Substituting K_0 in eq. 8 leads to the following relationship:

$$\frac{T_{*\delta}}{T_{*a}} = \frac{\delta K_{pyc}}{d_{pyc}K_0 + \delta K_{pyc}}. \quad (10)$$

In the above equation, apart from the effective thermal diffusivity at the IOI (eq. 22), K_0 , the remaining terms are insensitive to Γ . Hence $\frac{T_{*\delta}}{T_{*a}}$ takes similar values for all Γ for a given value of K_0 (Fig 3b). Thus, the efficiency of heat transfer at the IOI governs the BL thermal driving for BL-DSPs as long as it covaries with vertical mixing, underscoring the BL dynamics discussed earlier.

The manuscript discusses the revision of basal melt rate parameterizations under ice shelves, with a particular focus on cases where stable density stratification (pycnocline) suppresses vertical heat mixing. Using an Ice-Ocean Boundary Current Model (IOBCM), the authors analyze the dynamics of the boundary layer (BL) over a stable pycnocline and compare calculated melt rates with those parameterized in ocean models. The manuscript highlights that conventional parameterizations tend to overestimate melt rates under conditions of weak currents and strong stratification, as observed in the grounding zone of Thwaites Glacier (Davis et al, 2023).

The authors propose a new physics-based parameterization framework that considers the crucial role of limited diffusivity in the pycnocline and the balancing of heat fluxes. This approach aims to improve the representation of basal melting in low-resolution ocean models, contributing to a better understanding of ice shelf stability and sea level rise projections.

In general, I found the reading of the manuscript difficult. Considering the importance of the topic, I would have preferred a presentation of the results that was or more descriptive or with a better description of the many approximations and assumptions used to derive the equations. In a short paper (such this one), the contents and the passages, that sometimes are not intuitive, are hard to follow. The manuscript can certainly be considered for publication, but for this type of manuscript (compact), my suggestion is to move as much as possible the description of the derivation of the equations into the methods/supplementary material and then give more space to the description of the consequences and content of the figures. Indeed, the paper appears somehow very technical. Therefore, it may not be easily accessible to a reader who are not fully versed in the theoretical aspects of these processes. Thus, my main suggestion is to try to make the paper easier to read.

Minor points

Lines 109-111: are the values used for U_{ff} realistic for the ocean? Please explain why did you use this values. Furthermore, it could be useful to have an indication about how far from the IOI this velocity can be observed.

Line 148: is 30m a realistic estimate of the BL depth? It looks like that this value can be observed only in very high turbulent regimes.

Line 156: how did you chose the depth (30 m) of the BL?

Line 156: is it correct the figure indicated at the end of the line (Fig 1e)?

Line 159: is it correct to use the term spike? In my understanding, a spike is an anomalous value. Here it looks like that it is just and higher value.

Lines 185-186: considering that the typical depth of BL is less than 30 m according to the literature (see comment above for line 148) wasn't this consideration already somewhat expected?

Lines 228-229: In equation 10 the authors introduced K_0 but this is the first time they used it. In the method section it is indicated that it is the effective thermal diffusivity. It would be useful to indicate what it represents here too or to refer to the Methods equation 22.

Line 248: Is approximate the right verb to use here? In which sense $T_{*\delta}$ is approximated by u_{*0} ?

Line 263: You just introduced u_{*FF} assuming a linear profile for currents within the BL. Can you explain which could be the error in u_{*FF} ?

Lines 258 to 274: I suggest the authors to improve the clarity of this part of the manuscript.

Line 265: From what I see looking at figure 4c, the highest errors occur when $\Gamma = 0.006$ that is a value in the middle of the range used.

Line 299: What LES does mean?

Line 320: We have very little information and observations in the cavity, so the authors may be right when they say that temperature observations taken in a borehole may be close to those in the BL. I wonder what the distance from the BL at which T_w is measured and whether using different values the estimated \dot{m} has more realistic values.

Lines 340 to 342: A clearer explanation might help the reader to better understand this concept.

Comments on figures.

Figure 1 a,b,c i would draw the line representing the boundary layer depth with a different colour to avoid confusion with the curve representing the profiles of u , k and v in for the high far field velocity case.

Figure 2 : there are no indications in the figure caption about what the different colours of the dots represent. The legends in b4 and d4 are not enough.

Figure 3: please check the format of the letters used to identify the subplots. For instance, c) is not in bold.

Figure 4: please indicate for panel a what does the bold, continuous line represent. I guess the zero error line, but you should indicate it.

In the Dataset section Table 1, please explain what the parameter "Size" represents.

Review

Summary – Key results

This study presents a new formulation of a basal melt parameterisation for ice-ocean boundary layers to overcome current inconsistencies between parameterisation implemented in oceanic models and observations in relatively flat areas.

The revised parameterisation is based on the results of an Ice-Ocean Boundary Current Model (IOBCM) that parameterises turbulent processes using a mixing length approach. The authors first check that the model captures the desired physics.

Then, they compare the results of the model with the current parameterisation implemented in ocean models. Results suggest that when h (the depth of model first grid point) is greater than the depth of the boundary layer, the current parameterisation overestimates basal melting. Reducing the value of h could be a first step towards a more realistic estimate of melting in low-slope areas.

By assuming that warming by mixing across the pycnocline is the most important element, authors propose a new parameterisation framework and compare its performance with the IOBCM model. This parameterisation ensures that the observed physics in flat areas is properly represented, and seems relatively easy to use. The results are promising, although the parameterisation remains under development and needs to be further validated on more complex geometries.

Comment & suggestions

My overall impression of this paper is fairly positive. The paper sheds new light on ice-ocean interactions under ice shelves and proposes several ways of correcting a basal melt parameterisation in order to overcome inconsistencies between parameterisations and observations.

The methods are clearly explained and the figures are relevant for the reader's understanding.

This manuscript will likely be a significant contribution with revision and clarification about (i) recommendations for the modelling community and (ii) a wider discussion on the impact of the corrected process vs. the other processes neglected in the basal melt parameterisation.

I made a few comments. I split the comments into two categories: major (request) and minor (suggestion). The aim of this request is to ensure that the reader clearly understands what corrections can be implemented now in the ocean model and what sources of uncertainty remain and still need to be investigated.

Major points

- I think the abstract/conclusion might be clearer:
 - abstract: I think the authors should add in the abstract what the quick fix consists of, as it is one of the conclusions of the study. That's mean, reducing the distance between the first grid point and the base of the ice shelf.
 - conclusion: recommendations on the new formulation of the basal melt parameterisation are missing. Parameter values or indications on how to choose them should be provided and the remaining limitations specified. The authors focus on the calibration of the value of m_k , but it is not clear to me what values/range of values to consider for γ , K_b , C_d and d_{pyc} .
- I'm not an expert in basal melt parameterisation but I feel that a discussion about the impact of the results presented here is missing, and that the authors should attempt to put them into a more general context:
 - since the authors stress the importance of basal melting/geometry feedback, what is the validity of their new parameterisation in a context of changing geometry? (for example, are the value of calibrated parameters still valid in another geometry? for sloped IOI?)

- I understand that the generalisation to more complex and realistic geometry is planned for a future paper. Nevertheless, I think that a general discussion is missing on the absence of some processes that could also influence (in particular decrease) the melting under Thwaites.

For example:

- the impact of the tide in vertical mixing (Jourdain et al. 2019)
- the impact of accounting for heat conduction into the ice (Wiskandt et al. 2024)
- the impact of roughness like rift, crevasses (Poinelli et al. 2023, Schmidt et al. 2023)
- the impact of advection vs. vertical turbulent heat fluxes (Davis et al. 2025)

Minor points

L13: “basal melt rates” instead of “basal melts rates”

L14 and 54: “suppress” instead of “suppresses”

L21-22: sentence unclear. What are the quick fixes? I think the authors should add here what this quick fix consists of, as it is one of the conclusions of the study.

L52: The authors can also cite Schmidt et al. (2023) as the conclusion is similar.

L56-57: It would be interesting to calculate the slope under the ice shelves, using a dataset such as BedMachine for example, just to assess whether the low slopes under the ice shelves are common and whether many ice shelves could be affected by this overestimated melting.

Figure 1: Would it be possible to make the labels a little bigger or bold to make them easier to read?

I think the unit for the y-axis is missing for Fig 1g (D_{BL} in m?)

The authors could also be a little more precise about the legend in Figure 1a. HTC or PP is indicated for the red and blue curves. The authors could add an additional qualifier to more easily differentiate the difference between the two curve colours (like U_{ff} value?).

The choice of highlighting the depth of the boundary layer in the same colour as the profiles is a little confusing in Fig 1abc.

In Fig 1e, it is not clear that the authors are using two values for the C_d coefficient. Would it be possible to clarify this on the figure?

L150: Equation 25 is referred to several times in the body of the text. Wouldn't it be a good idea to introduce it in the body of the text?

Figure 2: It would be easier to understand if you put columns 1 and 3 next to each other and columns 2 and 4 next to each other, with each of the two columns having the same vertical amplitude for melt rate.

It would also be useful to remind the reader what m and m_h correspond to in the legend of the figure: “Comparison of calculated (m) and parameterised (m_h , BLM approach) melt rates...”

L222: sentence unclear. Which data are you talking about?

Figure 3: Why is the colour code changing between figures 3a and 3c? A single colour per gamma would be more appropriate.

L289-290: sentence unclear.

L292-294: the authors could specify that they are referring to figure 4ef for dots and lines.

Figure 5: As Georges VI and Amery are superimposed, it is not clear whether the parameterisation does not fit Amery or Georges V melt rate. What could explain the differences between the parameterisation and the boreholes values for Georges VI or Amery?

L343: The following paragraph is more a conclusion than a discussion. The previous part that focuses on the comparison of the parameterisation with observation and Yang's parameterisation is more similar to a discussion. Should the name of the paragraph be changed?

L464: extra parenthesis

Miscellaneous

Appropriate use of statistics and treatment of uncertainties: yes.

Conclusion: the conclusions and data interpretation are robust, valid and reliable.

References: yes

Clarity and context: the abstract is fairly clear and accessible (see comment on L21-22). Introduction and conclusions are appropriate.

Inflammatory material: no

References for this review (not cited in the paper)

Davis, P. E., Nicholls, K. W., Holland, D. M., Schmidt, B. E., Washam, P., Castro, B. F., ... & Makinson, K. (2025). Lateral Fluxes Drive Basal Melting Beneath Thwaites Eastern Ice Shelf, West Antarctica. *Geophysical Research Letters*, 52(3), e2024GL111873.

Jourdain, N. C., Molines, J. M., Le Sommer, J., Mathiot, P., Chanut, J., de Lavergne, C., & Madec, G. (2019). Simulating or prescribing the influence of tides on the Amundsen Sea ice shelves. *Ocean Modelling*, 133, 44-55.

Poinelli, M., Schodlok, M., Larour, E., Vizcaino, M., & Riva, R. (2023). Can rifts alter ocean dynamics beneath ice shelves?. *The Cryosphere*, 17(6), 2261-2283.

Schmidt, B. E., Washam, P., Davis, P. E., Nicholls, K. W., Holland, D. M., Lawrence, J. D., ... & Makinson, K. (2023). Heterogeneous melting near the Thwaites Glacier grounding line. *Nature*, 614(7948), 471-478.

Wiskandt, J., & Jourdain, N. (2024). Brief Communication: Representation of heat conduction into the ice in marine ice shelf melt modeling. *EGUsphere*, 2024, 1-10.

Revisiting Ice shelf Basal Melt Parameterisations for Ice-Ocean Boundary Layers over dynamically stable Pycnoclines

T. Jayasankar¹, A. Jenkins¹ *

¹University of Northumbria, Newcastle upon Tyne, United Kingdom.

Corresponding author: Adrian. Jenkins (adrian2.jenkins@northumbria.ac.uk)

Geography and Environmental Sciences

University of Northumbria, Newcastle upon Tyne, United Kingdom

Abstract

Accurate basal melt prediction under ice shelves is crucial for ice sheet stability assessments and sea level rise projections. A recent observational study in the eastern Thwaites Glacier grounding zone reported low basal melt rates despite warm ocean waters beneath the ice. Low current speeds and strong density stratification suppresses the vertical mixing of heat there, so the canonical model of ~~iceshelf~~ basal melting overestimates the melt rates. Hence, we revisit the basal melt parameterisation approach used in ocean models, applying an ice-ocean boundary current model to the simple case of a horizontal ice base. Such a setting yields a boundary layer over a dynamically stable pycnocline (BL-DSP). The pycnocline separates the cold, fresh and buoyant water within the boundary layer from the warmer ocean waters below and its low diffusivity limits the melt rate. Basal melt parameterisations that do not consider the impact of low pycnocline diffusivity can significantly overpredict melting for weaker far field currents. While we suggest quick fixes for this, we also propose a physics-based parameterisation framework for BL-DSPs that can emulate the physics in models as well as observations. The framework is expected to improve basal melting for BL-DSPs in low resolution ocean models.

Introduction

Floating ice shelves are an integral part of the climate system as they regulate the outflow from the Antarctic Ice Sheet. Their thinning can result in the acceleration of upstream ice flow and the retreat of grounding lines (Reese et al., 2018). Basal melting under ~~the~~ ice shelves is not only influenced by ocean forcing but also by the geometry of the ocean cavity beneath the ice (Holland et al., 2023; De Rydt & Naughten 2023). Considering that melting changes the geometry of ice shelves and the dependence of melting on geometry, accurate basal melting in coupled ice-sheet ocean models is important for capturing this important feedback mechanism.

The relationship between melting and geometry is primarily governed by the depth-dependence of the in-situ freezing point (T_f) of ocean water. The freezing point along ~~an Ice-Ocean Interface~~ (IOI) is usually expressed as a function of the depth (z) dependant pressure (P) and salinity (S) as follows, where λ_1 , λ_2 , and λ_3 are constants:

37

38 ~~Freezing point at IOI,~~ $T_f = \lambda_1 S + \lambda_2 + \lambda_3 P.$ (1)

[revised manuscript text omitted]

 **e), g) and h) as well.**

The quantitative differences between the schemes, as mentioned above, are caused by the
 respective diffusivities (K) and viscosities (ϑ), which are lower in the PP scheme than in HTC.
 Consequently, velocity gradients (Fig 1ab) in the HTC scheme are greater leading to higher values
 of friction velocity (u_{*0}) at the IOI (Fig 1e) for any given far field current (U_{FF}). Two trajectories
 of u_{*0} against U_{FF} are shown in Fig 1e for each scheme, corresponding to the two different values
 of $\sqrt{C_d}$ used (0.05 and 0.12) in the simulations, while the boundary layer thickness is determined
 almost uniquely by u_{*0} regardless of the chosen drag coefficient or mixing scheme (Fig. 1f).
 Finally, we look at the effective thermal driving ($T_{*\delta}$), which is the thermal driving at a small
 distance δ ($< 0.5\text{m}$) from the IOI, as this governs the melt rates at the IOI. In Fig 1gh, the thermal
 driving ratio, $T_{*\delta}/T_{*a}$, is plotted against $\log(u_{*0})$. **In both schemes, with an increase in u_{*0} , $T_{*\delta}$**
 **decreases due to the more effective transfer of heat to the IOI** (Eq 24 in Methods) that results from

[revised manuscript text omitted]

 shown as black lines in Fig 3a and closely trace the true profiles. The nonlinear reduction in $\frac{T_{*\delta}}{T_{*a}}$
 is quicker for higher values of Γ as it enables efficient cooling even for lower $T_{*\delta}$ to balance the
 warming that is primarily driven by u_{*0} , thus enabling quick development of a BL-DSP.
 Substituting, eq. 22 (Methods) in eq. 8 leads to the following relationship:

$$228 \quad \frac{T_{*\delta}}{T_{*a}} = \frac{\delta K_{pyc}}{d_{pyc}K_0 + \delta K_{pyc}}. \quad (10)$$

In the above equation, apart from K_0 , the remaining terms are insensitive to Γ . Hence $\frac{T_{*\delta}}{T_{*a}}$ takes
 similar values for all Γ for a given value of K_0 (Fig 3b). In Fig 3ab, there are some deviations from
 the profiles given by eq. 8 and eq. 10 respectively. These are cases where values of T_{*a} and Γ are
 extremely low, such that $(T_{*a} - T_{*\delta})$ in eq. 7 is not an acceptable representation of the T_* gradient
 within the pycnocline.

 **Figure 3: a) and b) Respective response of effective thermal driving to friction velocity (u_{*0})**
 **and to thermal diffusivity (K_0) at the IOI for a range of T_{*a} (0.005 to 4) and $\sqrt{C_d}$ (0.05 to**
 **0.12). c) Melt rates given by the IOBCM against u_{*0} for $T_{*a} = 1^\circ\text{C}$.**

Higher U_{FF} or u_{*0} leads to the condition $d_{pyc}u_{*0}\Gamma \gg K_{pyc}$. Ignoring K_{pyc} in eq. 8 yields:

$$T_{*δ} \cong \frac{K_{pyc}}{d_{pyc}u_{*0}\Gamma} T_{*a}. \quad (11)$$

Substituting this in eq. 25 (Methods), approximates \dot{m} as:

$$\dot{m} \cong \left(\frac{c}{L_i}\right) \frac{K_{pyc}}{d_{pyc}} T_{*a}. \quad (12)$$

Eq. 12 suggests that \dot{m} is independent of Γ . Fig 3c shows a plot of \dot{m} vs u_{*0} for different Γ and
 underscores the weak dependence of \dot{m} on Γ for a horizontal IOI at least in the IOBCM. This could
 be why ocean model studies on melt rate sensitivity to Γ (Mathiot et al., 2017; Jourdain et al.,
 2017) are inconclusive, and why the tuning of Γ to match basal \dot{m} for BL-DSPs may not always
 yield favourable outcomes. However, when $\frac{K_{pyc}}{d_{pyc}}$ is comparable to Γ , Γ becomes more relevant, as,
 for example, when instabilities associated with ~~sloped~~ ^{sloped} IOIs enhance vertical mixing.

Since, $T_{*\delta}$ can be approximated from u_{*0} (eq. 9 and 10), a parameterisation framework for \dot{m} can
 be built on parameterisation of u_{*0} . Assuming a linear profile for currents within the BL, a simple
 approximation for u_{*0} , u_{*FF} , can be expressed as follows:

$$251 \quad u_{*FF} = \frac{1}{2} |U_{FF}| \sqrt{C_d}, \quad (13)$$

Since u_{*FF} is used in further discussion, K_{pyc_FF} is introduced which is same as K_{pyc} but a function
 of u_{*FF} instead of u_{*0} :

$$255 \quad K_{pyc_FF} = K_b + m_{Kb} u_{*FF}, \quad (14)$$

$$256 \quad \dot{m} = \left(\frac{c}{L_i} \right) \left[\frac{K_{pyc_FF}}{d_{pyc} u_{*FF} \Gamma + K_{pyc_FF}} \right] u_{*FF} \Gamma T_{*a}. \quad (15)$$

At least for the range of $\sqrt{C_d}$ values used (0.05 to 0.12), u_{*FF} is a handy approximation of u_{*0} (Fig
 4a). In Fig 4b, predicted values of $T_{*\delta}$ (blue when the predictor is u_{*0} and red when it is u_{*FF}) are
 compared against the actual values given by the IOBCM. While blue markers are accurate, the red
 markers are mostly accurate when the predicted $T_{*\delta}$ is high (close to T_{*a}) or low (close to the
 limiting value of $T_{*\delta}$). The error is mostly in the region where $T_{*\delta}/T_{*a}$ quickly falls from 1 to its
 limiting value (Fig 3a). It is evident from Fig 4b that this error is due to the error in u_{*FF} . However,
 melt rate predictions using eq. 15 are not severely affected by this error except for extreme values
 of Γ (Fig 4c). One reason for this is that melt rates are proportional to the product of u_{*FF} and $T_{*\delta}$.
 By virtue of the inverse relationship between u_{*FF} and $T_{*\delta}$, any error in u_{*FF} will be at least
 partially cancelled by the consequent error in $T_{*\delta}$. Another reason is once $T_{*\delta}$ is close to its
 limiting value (which happens quickly), it remains more or less the same despite changes in u_{*FF} .
 Hence, if m_{Kb} is obtained by fitting predicted melt rates to data, errors will be confined to a regime
 of weaker currents, or lower u_{*0} . Since melting is also low for low u_{*0} , satisfactory prediction can
 be expected. Since, m_{Kb} governs warming by mixing across the pycnocline, it only affects the
 limiting value of $T_{*\delta}$, as illustrated by the sensitivity of $T_{*\delta}/T_{*a}$ to m_{Kb} for $\Gamma = 0.006$ (Fig 4d).
 Hence, just by tuning m_{Kb} , parametric models can be fitted to numerical models or observations
 for this special case of BL-DSP.

**Figure 4:** a) Comparison between friction velocity at the IOI given by the IOBCM (u_{*0}) and
 parameterised friction velocity (u_{*FF}) b) Comparison between $T_{*\delta}$ in the IOBCM and $T_{*\delta}$
 predicted when the predictors are u_{*FF} (red) and u_{*0} (blue). c) Comparison of melt rates
 given by the IOBCM and predicted melt rates using u_{*FF} as predictor for $T_{*\delta}$. The dotted
 lines mark 10% increments of the deviation from the zero-error line. d) $T_{*\delta}/T_{*a}$ vs u_{*0} for
 different values of m_{kb} . e) f) Parameterised \dot{m} for different values of m_{kb} and calculated \dot{m} vs U_{FF}
 for an ambient thermal driving of 1.5°C and 2.0°C respectively. Legends in c) and d) apply to e) and
 f) as well.

*Comparison with observation and other parameterisations*

We initially draw comparisons between the IOBCM and the Thwaites grounding zone
 observations (Davis et al., 2023), which motivated the study. The \dot{m} versus U_{FF} for cases where
 $T_{*a} = 1.5$ and $T_{*a} = 2.0$ (which span the observed range 1.66 - 1.94°C) are presented in Fig 4e
 and 4f respectively. Currents ranging from 2.0 to 5.4 cm/s were observed under the ice shelf,
 yielding conventionally calculated \dot{m} of 14.0 to 30.0 m/y versus observed rates of 2.0 to 5.0 m/y.
 Davis et al., (2023) reasoned that the suppressed basal melting resulted from stratification and
 weaker currents. While observed melt rates are between 2.0 and 5.0 m/yr, melt rates given by both
 the IOBCM (dots) and the parametrised IOBCM (black lines) are conservative with a maximum
 of 1 m/yr. However, if m_{Kb} is increased to 0.025 in the parameterised model, \dot{m} can be increased
 to the ballpark of observed melting (red line). This case is presented to show the versatility of this
 parameterisation framework for the case of a BL-DSP.

We next draw comparisons with a recent study by Yung et al. (2024 hereafter Y24), which
 followed an alternative approach to the parameterisation of basal melting for the cases when
 mixing is suppressed by stratification, adopting a functional form for Γ based on LES. In the
 reduced form of the parameterised Γ_T and Γ_S (Rosevear et al., 2022; Y24), Γ_{para} can be expressed
 as:

$$302 \quad \Gamma_{para} = \frac{K \frac{\partial T_*}{\partial z}}{u_{*0} T_{*a}}. \quad (16)$$

The variation of Γ_{para} with respect to L^+ (the ratio of Obukhov to viscous length scales, which is
 proportional to u_{*0}^{-4}) is similar but inverse in nature (see Fig.1 of Y24) to the variation of $T_{*\delta}$
 with respect to $\log(u_{*0})$ shown in Fig 3a. The variation of Γ_{para} and $T_{*\delta}$ are only observed for
 lower U_{FF} and they plateau thereafter. Hence, in Y24, Γ_{para} is parameterised as an increasing
 function of L^+ until it reaches a cutoff value $\Gamma_{para,CC}$ and remains constant thereafter. The
 cutoff/threshold value ($\Gamma_{T,CC}$ and $\Gamma_{S,CC}$ in Y24) is presented as an upper limit for the stratification
 feedback. A similar approach in the current study would result in fixing a lower cutoff value for
 $T_{*\delta}$ after a certain value for u_{*0} and fitting an equation for its reduction in the low u_{*0} or U_{FF}
 regime. However, by conserving heat fluxes into the BL in eq. 7 and focussing on $T_{*\delta}$, this
 variation and the associated physics are naturally preserved in eq. 15 without introducing
 cutoffs/thresholds.

Fig. 5 shows a comparison of the parameterisations of Y24 and eq. 15 against observations (Table
 B1 of Y24). With $m_{Kb} = 0.025$, the \dot{m} in eq. 15 is lower than Y24 and the observed melt rates.
 For $U_{FF} \geq 0.1 m/s$ Y24 overestimates \dot{m} while eq. 15 underpredicts the same. However, a
 significantly high m_{Kb} of 10 in eq. 15 yields \dot{m} that matches well with observation. Such a high
 value for m_{Kb} implies that, no matter what, $T_{*\delta}$ always remains close to T_{*a} . Also, any further
 increase in m_{Kb} has little to no effect on $T_{*\delta}$ and \dot{m} , as the maximum possible value for $T_{*\delta}$ is T_{*a} .
 Hence, in this case we suspect that the T_{*a} sampled by the borehole observations could be closer
 to values inside BL than to that of far-field ocean. Also, such a high m_{Kb} did not overestimate \dot{m}
 for low U_{FF} . This is because, no matter what, the equations constrain the value of $T_{*\delta}$ between 0
 and T_{*a} .

**Figure 5: Comparison between borehole observations, basal melt parameterisation by Yung**
 **et al., (2024) and current parameterisation framework for different far field velocities (U_{FF}).**
 **The data are based on melt rates, U_{FF} and far field ambient thermal driving compiled by**
 **Yung et al., (2024).**

Clearly, both the parameterisations of Y24 and eq. 15 have some merit. The former is built on a
 much more complete physical model, while the simplicity of the latter allows for flexible tuning.
 They are, however, quite different in concept. Y24, in common with other recent studies, focuses
 on improving the melt rate computed from eq. 25 (Methods) through a careful consideration of the
 boundary layer physics subsumed into the turbulent transfer coefficient, Γ . In contrast, eq. 15
 exploits the principle that for a **horizontal IOI, lateral supply of heat must be negligible**, so $T_{*\delta}$
 will adjust until vertical heat fluxes are balanced, regardless of the value of Γ . Heat flux through
 the pycnocline will then be the controlling factor, since diffusivity beyond the boundary layer will
 always be lower than within it. This represents a redirection of primary focus from the BL and the
 IOI to the pycnocline and may be more widely applicable than the simple cases investigated in this
 study. Burchard et al. (2022) found that high resolution in the region of the pycnocline was critical

to the performance of their model of the ice-shelf-ocean boundary layer, regardless of resolution
 elsewhere, a finding that also underscores the heat flux across the pycnocline as being key.

**Discussion**

We have investigated the current basal melt parameterisation approach in z-coordinate ocean
 models for a stratified water column yielding a boundary layer (BL) over a dynamically stable
 pycnocline. The work was motivated by a recent study (Davis et al., 2023) that reported that the
 conventional model of **iceshelf** basal melting overestimates the observed basal melting (2.0-5.0
 348 m/yr) under eastern Thwaites glacier significantly (as high as 14.0 to 30.0 m/yr). It has been
 reasoned that weak mixing due to low current speed and strong density stratification under the ice-
 ocean interface (IOI) suppresses the heat transport from warm ambient ocean beneath to the IOI.
 Hence, we revisited the current basal melt parameterisation in ocean models for the simple case of
 a horizontal IOI, yielding a BL over a dynamically stable pycnocline (BL-DSP).

[revised manuscript text omitted]

411

412 Boundary conditions are as follows.

$$413 \quad \text{At the IOI where } z = 0: \quad u = 0, \quad v = 0, \quad T_* = 0$$

$$414 \quad \text{In the far field where } z = -\infty: \quad u = u_{FF}, \quad v = v_{FF}, \quad T_* = T_{*a}$$

$$415 \quad T_{*a} = T_a - (\lambda_1 S_a + \lambda_2 + \lambda_3 Z)$$

The equations above are solved for variables u , v and T_* as functions of time t along the z axis. T_{*a}
 here represents thermal driving derived from ambient ocean temperature (T_a) and salinity (S_a).
 The motivation for combining salinity and temperature to derive T_* is detailed in Jenkins et al.
 (2010). Using the thermal driving as a variable yields a simple boundary condition at the IOI of 0.
 Velocities in x (North-South) and y (East-West) directions along the IOI are denoted $u(z)$ and $v(z)$
 respectively.

In Eqs. 17 and 18, the first terms on the RHS represent the acceleration in the x and y directions
 respectively due to geostrophic far field currents (u_{FF} and v_{FF} respectively) that are driven by the
 deviation of the ice shelf from its level of free floatation and deflected by the Coriolis effect. The
 second terms are viscous resistance. An additional term to account for density driven currents
 along a sloped IOI presented in Jenkins' (2016, 2021) version of (7) is omitted here as our study
 is limited to flat IOIs. For convenience, the forcing on the system is expressed here as $f v_{FF}$ and
 $f u_{FF}$ where f is the local Coriolis parameter. In Jenkins (2016, 2012) those terms are written
 explicitly as $g \cos \alpha \frac{\partial \eta}{\partial x}$ and $g \cos \alpha \frac{\partial \eta}{\partial y}$ respectively, where η is the vertical deviation of the IOI
 from its equilibrium position. In Eq 19, the profile for T_* is determined solely by the eddy
 diffusivity, K which is dependent on u , v and T_* itself.

The LTC is a mixing length model, in which both ϑ and K are linear functions of mixing length
 and friction velocity, u_* (Jenkins 2021). The mixing length grows with distance from the IOI,
 reaching a limiting value that is set by stratification and Earth rotation, while u_* is computed as
 follows:

$$436 \quad \text{Friction velocity, } u_*(z) = \left(\vartheta \left| \frac{\partial \mathbf{u}(z)}{\partial z} \right| \right)^{\frac{1}{2}}. \quad (20)$$

*Ice Ocean Interface*

Based on observations in turbulent boundary layer flows, the effective thermal diffusivity (K_0) and
 viscosity (ϑ_0) in the vicinity of a boundary or wall are expressed as linear functions of u_* at the
 IOI (u_{*0}) as follows (Jenkins 2021):

$$441 \quad \text{Friction velocity at IOI, } u_{*0} = \left(\vartheta_0 \left| \frac{\mathbf{u}_\delta}{\delta} \right| \right)^{\frac{1}{2}}, \quad (21)$$

$$442 \quad K_0 = u_{*0} \Gamma \delta, \quad (22)$$

$$443 \quad \vartheta_0 = u_{*0} \sqrt{C_d} \delta. \quad (23)$$

The subscripts in the above equations represent model grid nodes, where 0 represents the IOI and
 δ the first grid point below, a small distance ($\delta < 0.5\text{m}$) away from the IOI. From eq. 22 and 19,
 the heat flux at the IOI can be expressed as:

$$447 \quad q = u_{*0} \Gamma T_{*\delta}, \quad (24)$$

leading to an expression for melt rate at the IOI given by:

$$449 \quad \dot{m} = \left(\frac{c}{L_i} \right) u_{*0} \Gamma T_{*\delta}. \quad (25)$$

Eq 25 and 1 are similar except that in eq. 25, u_{*0} replaces u_{*h} and $T_{*\delta}$ replaces T_{*h} . In practice,
 u_{*0} and $T_{*\delta}$ are coupled, and the BL dynamics governs the coupling.

*Datasets*

Datasets are created by running a range of IOBCM simulations with varying model parameters.
 The parameter space of the datasets consists of ambient thermal driving (T_{*a}), turbulent transfer
 coefficient (Γ), drag coefficient ($\sqrt{C_d}$), far-field current speed (U_{FF}) and Coriolis parameter (f).
 Two separate datasets (Table 1) are created for this manuscript. The simulations are run until the
 model assumes a steady state (i.e., up to 90 inertial cycles). The datasets are created by time
 averaging model variables over the last 10 (81st to 90th) inertial cycles.

**Table 1: Details of datasets generated using the IOBCM**

	Dataset-1	Dataset-2
Purpose	Studying boundary layer dynamics in the HTC and PP mixing schemes	For parameterisation framework using the HTC mixing scheme
Size	480 (240 for each mixing scheme)	3544
T_{*a} ($^{\circ}\text{C}$)	1 and 2	0.005, 0.01, 0.02, 0.03, 0.04, 0.05, 0.07, 0.10, 0.15, 0.20, 0.25, 0.50, 1.0, 1.5, 2.0, 2.5, and 4.0
Γ	0.003, 0.006, 0.009, 0.012, and 0.1	0.001, 0.003, 0.006, 0.009, 0.012, and 0.1
$\sqrt{C_d}$	0.05 and 0.12	0.05, 0.055, 0.079 and 0.12
U_{FF} (m/s)	0.0014 to 1.05	$6.7541 \cdot 10^{-6}$ to 1.48
f (rads/s)	$-1.408 \cdot 10^{-4}$	$-1.452 \cdot 10^{-4}$, $-1.408 \cdot 10^{-4}$, $-1.321 \cdot 10^{-4}$, and $-1.194 \cdot 10^{-4}$

**Acknowledgments**

This research is a part of the PROTECT (Projecting Sea-Level Rise: from Ice Sheets to Local
 Implications (protect-slr.eu)) project supported by funding from the European Union's Horizon
 2020 research and innovation program.

**Author contributions**

467 T.J and A.J. conceived the study. T.J. undertook the model runs, developed the parameterisation
 and led the writing the paper. A.J. assisted with the interpretation of the data and writing of the
 paper.

**Competing interests**

The authors declare no competing interests.

**Materials & Correspondence**

Correspondence and material requests should be addressed to Adrian Jenkins
(adrian2.jenkins@northumbria.ac.uk).

**Data and Code Availability**

The data used in this study is generated solving the model detailed in Jenkins (2021) using Matlab.
The model is available at <https://doi.org/10.5281/zenodo.13381662>. The generated datasets and codes to
reproduce the figures in this study are available at <https://doi.org/10.5281/zenodo.14628475>.

All variables in this manuscript are detailed in Jenkins (2021,
[https://researchportal.northumbria.ac.uk/en/publications/shear-stability-and-mixing-within-the-](https://researchportal.northumbria.ac.uk/en/publications/shear-stability-and-mixing-within-the-ice-shelf-ocean-boundary-cu)
[ice-shelf-ocean-boundary-cu](https://researchportal.northumbria.ac.uk/en/publications/shear-stability-and-mixing-within-the-ice-shelf-ocean-boundary-cu)), with their respective values given in Table 1.

**References**

- Burchard, H., Bolding, K., Jenkins, A., Losch, M., Reinert, M., & Umlauf, L. (2022). The
vertical structure and entrainment of subglacial melt water plumes. *Journal of Advances*
*in Modeling Earth Systems*, *14*, e2021MS002925.
- Davis, P. E., Nicholls, K. W., Holland, D. M., Schmidt, B. E., Washam, P., Riverman, K. L., . . .
others. (2023). Suppressed basal melting in the eastern Thwaites Glacier grounding zone.
*Nature*, *614*, 479–485.
- De Rydt, J., & Naughten, K. (2023). Geometric amplification and suppression of ice-shelf basal
melt in West Antarctica. *EGU sphere*, *2023*, 1–36.
- Favier, L., Jourdain, N. C., Jenkins, A., Merino, N., Durand, G., Gagliardini, O., . . . Mathiot, P.
(2019). Assessment of sub-shelf melting parameterisations using the ocean–ice-sheet
coupled model NEMO (v3. 6)–Elmer/Ice (v8. 3). *Geoscientific Model Development*, *12*,
2255–2283.
- Holland, D. M., & Jenkins, A. (1999). Modeling thermodynamic ice–ocean interactions at the
base of an ice shelf. *Journal of physical oceanography*, *29*, 1787–1800.
- Holland, P. R., Bevan, S. L., & Luckman, A. J. (2023). Strong ocean melting feedback during the
recent retreat of Thwaites Glacier. *Geophysical Research Letters*, *50*,
e2023GL103088.
- Jenkins, A. (2016). A simple model of the ice shelf–ocean boundary
layer and current. *Journal of Physical Oceanography*, *46*, 1785–1803.
- Jenkins, A. (2021). Shear, stability, and mixing within the ice shelf–ocean boundary current.
*Journal of Physical Oceanography*, *51*, 2129–2148.
- Jenkins, A., Nicholls, K. W., & Corr, H. F. (2010). Observation and parameterization of ablation
at the base of Ronne Ice Shelf, Antarctica. *Journal of Physical Oceanography*, *40*, 2298–
2312.
- Jourdain, N. C., Mathiot, P., Merino, N., Durand, G., Le Sommer, J., Spence, P., . . . Madec, G.
(2017). Ocean circulation and sea-ice thinning induced by melting ice shelves in the A
mundsen Sea. *Journal of Geophysical Research: Oceans*, *122*, 2550–2573.

- Losch, M. (2008). Modeling ice shelf cavities in az coordinate ocean general circulation model.
*Journal of Geophysical Research: Oceans*, 113.
- Mathiot, P., Jenkins, A., Harris, C., & Madec, G. (2017). Explicit representation and
parametrised impacts of under ice shelf seas in the z* coordinate ocean model NEMO
3.6, *Geosci. Model Dev.*, 10, 2849–2874. *Explicit representation and parametrised*
*impacts of under ice shelf seas in the z* coordinate ocean model NEMO 3.6, Geosci.*
*Model Dev.*, 10, 2849–2874.
- McPhee, M. (2008). *Air-ice-ocean interaction: Turbulent ocean boundary layer exchange*
*processes*. Springer Science & Business Media.
- McPhee, M. G. (1999). Parameterization of mixing in the ocean boundary layer. *Journal of*
*marine systems*, 21, 55–65.
- Reed, B., Green, J. M., Jenkins, A., & Gudmundsson, G. H. (2024). Recent irreversible retreat
phase of Pine Island Glacier. *Nature Climate Change*, 14, 75–81.
- Reese, R., Gudmundsson, G. H., Levermann, A., & Winkelmann, R. (2018). The far reach of ice-
shelf thinning in Antarctica. *Nature Climate Change*, 8, 53–57.
- Rosevear, M. G., Gayen, B., & Galton-Fenzi, B. K. (2022). Regimes and transitions in the basal
melting of Antarctic ice shelves. *Journal of Physical Oceanography*, 52, 2589–2608.
- Yung, C. K., Rosevear, M. G., Morrison, A. K., Hogg, A. M., & Nakayama, Y. (2024). Stratified
suppression of turbulence in an ice shelf basal melt parameterisation. *EGUsphere*, 2024,
1–43.

Revisiting Ice shelf Basal Melt Parameterisations for Ice-Ocean Boundary Layers over dynamically stable Pycnoclines

T. Jayasankar^{1,2}, A. Jenkins^{1*}.

¹University of Northumbria, Newcastle upon Tyne, United Kingdom.

²Now at Norwegian Polar Institute Fram Centre, Tromsø, Norway.

Corresponding author: Adrian. Jenkins (adrian2.jenkins@northumbria.ac.uk)

Geography and Environmental Sciences

University of Northumbria, Newcastle upon Tyne, United Kingdom.

Abstract

Accurate basal melt prediction under ice shelves is crucial for ice sheet stability assessments and sea level rise projections. Recent observations from the eastern Thwaites Glacier grounding zone reported low basal melt rates despite warm ocean waters beneath the ice. Low current speeds and strong density stratification suppress the vertical mixing of heat there, so the canonical model of ice shelf basal melting overestimates the melt rates. Hence, we revisit the basal melt parameterisation approach used in ocean models, applying an ice-ocean boundary current model to the simple case where a horizontal ice base yields a boundary layer over a dynamically stable pycnocline (BL-DSP). The pycnocline separates the cold, fresh and buoyant water within the boundary layer from the warmer ocean waters below and its low diffusivity limits the melt rate. Basal melt parameterisations that do not consider the impact of low pycnocline diffusivity can significantly overpredict melting for weaker far field currents. We suggest a quick fix for this (prescribing low melt rates for slower currents) but also propose a physics-based parameterisation framework for BL-DSPs that can emulate the physics in models and observations. The framework is expected to improve basal melting for BL-DSPs in low resolution ocean models.

Introduction

[revised manuscript text omitted]

 Davis et al., 2023) versus observed rates of 2.0 to 5.0 m/y. Davis et al., (2023) reasoned that the
 suppressed basal melting resulted from stratification and weaker currents. While observed melt
 rates are between 2.0 and 5.0 m/yr, melt rates given by both the IOBCM (Fig 4ef: dots) and the
 parametrised IOBCM (Fig 4ef: black lines) are conservative with a maximum of 1 m/yr. However,
 if m_{Kb} is increased to 0.025 in the parameterised model, \dot{m} can be increased to the ballpark of
 observed melting (red line), tuning that might account for the ice shelf slope. For a general
 parameterisation using eq. 15, instead of using constant values for m_{Kb} and d_{pyc} , they should be
 expressed as a function of IOI slope. The basic case for flat ice shelves is presented to show the
 versatility of this parameterisation framework for the case of a BL-DSP.

We next draw comparisons with a recent study by Yung et al. (2024 hereafter Y24), which
 followed an alternative approach to the parameterisation of basal melting for cases when mixing
 is suppressed by stratification, adopting a functional form for Γ based on Large Eddy Simulations.
 In the reduced form of the parameterised Γ_T and Γ_S (Rosevear et al., 2022; Y24), Γ_{para} can be
 expressed as:

[revised manuscript text omitted]

489

490 Boundary conditions are as follows.

$$491 \quad \text{At the IOI where } z = 0: \quad u = 0, \quad v = 0, \quad T_* = 0$$

$$492 \quad \text{In the far field where } z = -\infty: \quad u = u_{FF}, \quad v = v_{FF}, \quad T_* = T_{*a}$$

$$493 \quad T_{*a} = T_a - (\lambda_1 S_a + \lambda_2 + \lambda_3 Z)$$

494 The equations above are solved for variables u , v and T_* as functions of time t along the z axis. T_{*a}
 495 here represents thermal driving derived from ambient ocean temperature (T_a) and salinity (S_a).
 496 The motivation for combining salinity and temperature to derive T_* is detailed in Jenkins et al.

[revised manuscript text omitted]

**Materials & Correspondence**

Correspondence and material requests should be addressed to Adrian Jenkins
(adrian2.jenkins@northumbria.ac.uk).

**Data and Code Availability**

The data used in this study is generated solving the model detailed in Jenkins (2021) using Matlab.
The model is available at <https://doi.org/10.5281/zenodo.13381662>. The generated datasets and codes to
reproduce the figures in this study are available at <https://doi.org/10.5281/zenodo.14628475>.

All variables in this manuscript are detailed in Jenkins (2021),
[https://researchportal.northumbria.ac.uk/en/publications/shear-stability-and-mixing-within-the-](https://researchportal.northumbria.ac.uk/en/publications/shear-stability-and-mixing-within-the-ice-shelf-ocean-boundary-cu)
[ice-shelf-ocean-boundary-cu](https://researchportal.northumbria.ac.uk/en/publications/shear-stability-and-mixing-within-the-ice-shelf-ocean-boundary-cu)), with their respective values given in Table 1.

**References**

- Burchard, H., Bolding, K., Jenkins, A., Losch, M., Reinert, M., & Umlauf, L. (2022). The
vertical structure and entrainment of subglacial melt water plumes. *Journal of Advances*
*in Modeling Earth Systems*, *14*, e2021MS002925.
- Davis, P. E., Nicholls, K. W., Holland, D. M., Schmidt, B. E., Washam, P., Riverman, K. L., . . .
others. (2023). Suppressed basal melting in the eastern Thwaites Glacier grounding zone.
*Nature*, *614*, 479–485.
- Davis, P. E., Nicholls, K. W., Holland, D. M., Schmidt, B. E., Washam, P., Castro, B. F., ... &
Makinson, K. (2025). Lateral Fluxes Drive Basal Melting Beneath Thwaites Eastern Ice
Shelf, West Antarctica. *Geophysical Research Letters*, *52*(3), e2024GL111873.
- De Rydt, J., & Naughten, K. (2023). Geometric amplification and suppression of ice-shelf basal
melt in West Antarctica. *EGU sphere*, *2023*, 1–36.
- Favier, L., Jourdain, N. C., Jenkins, A., Merino, N., Durand, G., Gagliardini, O., . . . Mathiot, P.
(2019). Assessment of sub-shelf melting parameterisations using the ocean–ice-sheet
coupled model NEMO (v3. 6)–Elmer/Ice (v8. 3). *Geoscientific Model Development*, *12*,
2255–2283.
- Holland, D. M., & Jenkins, A. (1999). Modeling thermodynamic ice–ocean interactions at the
base of an ice shelf. *Journal of physical oceanography*, *29*, 1787–1800.
- Holland, P. R., Bevan, S. L., & Luckman, A. J. (2023). Strong ocean melting feedback during the
recent retreat of Thwaites Glacier. *Geophysical Research Letters*, *50*,
e2023GL103088.
- Jenkins, A. (2016). A simple model of the ice shelf–ocean boundary
layer and current. *Journal of Physical Oceanography*, *46*, 1785–1803.
- Jenkins, A. (2021). Shear, stability, and mixing within the ice shelf–ocean boundary current.
*Journal of Physical Oceanography*, *51*, 2129–2148.
- Jenkins, A., Nicholls, K. W., & Corr, H. F. (2010). Observation and parameterization of ablation
at the base of Ronne Ice Shelf, Antarctica. *Journal of Physical Oceanography*, *40*, 2298–
2312.

- Jourdain, N. C., Mathiot, P., Merino, N., Durand, G., Le Sommer, J., Spence, P., . . . Madec, G.
(2017). Ocean circulation and sea-ice thinning induced by melting ice shelves in the A
mundsen Sea. *Journal of Geophysical Research: Oceans*, *122*, 2550–2573.
- Jourdain, N. C., Molines, J. M., Le Sommer, J., Mathiot, P., Chanut, J., de Lavergne, C., &
Madec, G. (2019). Simulating or prescribing the influence of tides on the Amundsen Sea
ice shelves. *Ocean Modelling*, *133*, 44–55.
- Losch, M. (2008). Modeling ice shelf cavities in a z-coordinate ocean general circulation model.
*Journal of Geophysical Research: Oceans*, *113*.
- Mathiot, P., Jenkins, A., Harris, C., & Madec, G. (2017). Explicit representation and
parametrised impacts of under ice shelf seas in the z* coordinate ocean model NEMO
3.6. *Geosci. Model Dev.*, *10*, 2849–2874. *Explicit representation and parametrised*
*impacts of under ice shelf seas in the z* coordinate ocean model NEMO 3.6, Geosci.*
*Model Dev.*, *10*, 2849–2874.
- McPhee, M. (2008). *Air-ice-ocean interaction: Turbulent ocean boundary layer exchange*
*processes*. Springer Science & Business Media.
- McPhee, M. G. (1999). Parameterization of mixing in the ocean boundary layer. *Journal of*
*marine systems*, *21*, 55–65.
- Reed, B., Green, J. M., Jenkins, A., & Gudmundsson, G. H. (2024). Recent irreversible retreat
phase of Pine Island Glacier. *Nature Climate Change*, *14*, 75–81.
- Reese, R., Gudmundsson, G. H., Levermann, A., & Winkelmann, R. (2018). The far reach of ice-
shelf thinning in Antarctica. *Nature Climate Change*, *8*, 53–57.
- Rosevear, M. G., Gayen, B., & Galton-Fenzi, B. K. (2022). Regimes and transitions in the basal
melting of Antarctic ice shelves. *Journal of Physical Oceanography*, *52*, 2589–2608.
- Schmidt, B. E., Washam, P., Davis, P. E., Nicholls, K. W., Holland, D. M., Lawrence, J. D., ... &
Makinson, K. (2023). Heterogeneous melting near the Thwaites Glacier grounding line.
*Nature*, *614*(7948), 471–478.
- Wiskandt, J., & Jourdain, N. (2024). Brief Communication: Representation of heat conduction
into the ice in marine ice shelf melt modeling. *EGUsphere*, 2024, 1–10.
- Yung, C. K., Rosevear, M. G., Morrison, A. K., Hogg, A. M., & Nakayama, Y. (2024). Stratified
suppression of turbulence in an ice shelf basal melt parameterisation. *EGUsphere*, 2024,
1–43.